# Beyond SNP heritability: Polygenicity and discoverability of phenotypes estimated with a univariate Gaussian mixture model

Dominic Holland[1,2]*, Oleksandr Frei[3], Rahul Desikan[4†], Chun-Chieh Fan[1,5,6], Alexey A. Shadrin[3], Olav B. Smeland[3,7], V. S. Sundar[1,5], Paul Thompson[8], Ole A. Andreassen[3,7], Anders M. Dale[1,2,5,9]

**1** Center for Multimodal Imaging and Genetics, University of California at San Diego, La Jolla, California, United States of America, **2** Department of Neurosciences, University of California, San Diego, La Jolla, California, United States of America, **3** NORMENT, KG Jebsen Centre for Psychosis Research, Institute of Clinical Medicine, University of Oslo, Oslo, Norway, **4** Department of Radiology, University of California, San Francisco, San Francisco, California, United States of America, **5** Department of Radiology, University of California, San Diego, La Jolla, California, United States of America, **6** Department of Cognitive Sciences, University of California at San Diego, La Jolla, California, United States of America, **7** Division of Mental Health and Addiction, Oslo University Hospital, Oslo, Norway, **8** Keck School of Medicine, University of Southern California, Los Angeles, California, United States of America, **9** Department of Psychiatry, University of California, San Diego, La Jolla, California, United States of America

† Deceased.
* dominicholland@gmail.com

**Data Availability Statement:** All relevant data are within the manuscript and its Supporting Information files.

## Abstract

Estimating the polygenicity (proportion of causally associated single nucleotide polymorphisms (SNPs)) and discoverability (effect size variance) of causal SNPs for human traits is currently of considerable interest. SNP-heritability is proportional to the product of these quantities. We present a basic model, using detailed linkage disequilibrium structure from a reference panel of 11 million SNPs, to estimate these quantities from genome-wide association studies (GWAS) summary statistics. We apply the model to diverse phenotypes and validate the implementation with simulations. We find model polygenicities (as a fraction of the reference panel) ranging from $\simeq 2 \times 10^{-5}$ to $\simeq 4 \times 10^{-3}$, with discoverabilities similarly ranging over two orders of magnitude. A power analysis allows us to estimate the proportions of phenotypic variance explained additively by causal SNPs reaching genome-wide significance at current sample sizes, and map out sample sizes required to explain larger portions of additive SNP heritability. The model also allows for estimating residual inflation (or deflation from over-correcting of z-scores), and assessing compatibility of replication and discovery GWAS summary statistics.

## Author summary

There are ~10 million common variants in the genome of humans with European ancestry. For any particular phenotype a number of these variants will have some causal effect.

**Funding:** The author(s) received no specific funding for this work.

**Competing interests:** Dr. Dale is a founder of and holds equity in CorTechs Labs, Inc, and serves on its Scientific Advisory Board. He is a member of the Scientific Advisory Board of Human Longevity, Inc. and receives funding through research agreements with General Electric Healthcare and Medtronic, Inc. The terms of these arrangements have been reviewed and approved by UCSD in accordance with its conflict of interest policies. Dr. Andreassen is a consultant for HealthLytix. The remaining authors have no competing interest.

It is of great interest to be able to quantify the number of these causal variants and the strength of their effect on the phenotype.

Genome wide association studies (GWAS) produce very noisy summary statistics for the association between subsets of common variants and phenotypes. For any phenotype, these statistics collectively are difficult to interpret, but buried within them is the true landscape of causal effects. In this work, we posit a probability distribution for the causal effects, and assess its validity using simulations. Using a detailed reference panel of $\simeq 11$ million common variants – among which only a small fraction are likely to be causal, but allowing for non-causal variants to show an association with the phenotype due to correlation with causal variants—we implement an exact procedure for estimating the number of causal variants and their mean strength of association with the phenotype. We find that, across different phenotypes, both these quantities—whose product allows for lower bound estimates of heritability—vary by orders of magnitude.

## Introduction

The genetic components of complex human traits and diseases arise from hundreds to likely many thousands of single nucleotide polymorphisms (SNPs) [1], most of which have weak effects. As sample sizes increase, more of the associated SNPs are identifiable (they reach genome-wide significance), though power for discovery varies widely across phenotypes. Of particular interest are estimating the proportion of common SNPs from a reference panel (polygenicity) involved in any particular phenotype; their effective strength of association (discoverability, or causal effect size variance); the proportion of variation in susceptibility, or phenotypic variation, captured additively by all common causal SNPs (approximately, the narrow sense heritability), and the fraction of that captured by genome-wide significant SNPs—all of which are active areas of research [2–9]. The effects of population structure [10], combined with high polygenicity and linkage disequilibrium (LD), leading to spurious degrees of SNP association, or inflation, considerably complicate matters, and are also areas of much focus [11–13]. Despite these challenges, there have been recent significant advances in the development of mathematical models of polygenic architecture based on GWAS [14, 15]. One of the advantages of these models is that they can be used for power estimation in human phenotypes, enabling prediction of the capabilities of future GWAS.

Here, in a unified approach explicitly taking into account LD, we present a model relying on genome-wide association studies (GWAS) summary statistics (z-scores for SNP associations with a phenotype [16]) to estimate polygenicity ($\pi_1$, the proportion of causal variants in the underlying reference panel of approximately 11 million SNPs from a sample size of 503) and discoverability ($\sigma_\beta^2$, the causal effect size variance), as well as elevation of z-scores due to any residual inflation of the z-scores arising from variance distortion ($\sigma_0^2$, which for example can be induced by cryptic relatedness), which remains a concern in large-scale studies [10]. We estimate $\pi_1$, $\sigma_\beta^2$, and $\sigma_0^2$, by postulating a z-score probability distribution function (pdf) that explicitly depends on them, and fitting it to the actual distribution of GWAS z-scores.

Estimates of polygenicity and discoverability allow one to estimate compound quantities, like narrow-sense heritability captured by the SNPs [17]; to predict the power of larger-scale GWAS to discover genome-wide significant loci; and to understand why some phenotypes have higher power for SNP discovery and proportion of heritability explained than other phenotypes.

In previous work [18] we presented a related model that treated the overall effects of LD on z-scores in an approximate way. Here we take the details of LD explicitly into consideration, resulting in a conceptually more basic model to predict the distribution of z-scores. We apply the model to multiple phenotype datasets, in each case estimating the three model parameters and auxiliary quantities, including the overall inflation factor λ, (traditionally referred to as genomic control [19]) and narrow sense heritability, $h^2$. We also perform extensive simulations on genotypes with realistic LD structure in order to validate the interpretation of the model parameters. A discussion of the relation of the present paper to other work is provided in the first section of the S1 Appendix (pp. S2-S3).

## Materials and methods

### Overview

Our basic model is a simple postulate for the distribution of causal effects (denoted $\beta$ below) [20]. Our model assumes that only a fraction of all SNPs are in some sense causally related to any given phenotype. We work with a reference panel of approximately 11 million SNPs with 503 samples, and assume that all common causal SNPs (minor allele frequency (MAF) > 0.002) are contained in it. Any given GWAS will have z-scores for a subset of these reference SNPs (we use the term "typed" below to refer to GWAS SNPs with z-scores, whether they were directly genotyped or their genotype was imputed). When a z-score partially involves a latent causal component (i.e., not pure noise), we assume that it arises through LD with neighboring causal SNPs, or that it itself is causal.

We construct a pdf for z-scores that directly follows from the underlying distribution of effects. For any given typed SNP's z-score, it is dependent on the other SNPs the focal SNP is in LD with (SNPs that are "tagged" by the focal SNP), taking into account their LD with the focal SNP and their heterozygosity (i.e., it depends not just on the focal typed SNP's total LD and heterozygosity, but also on the distribution of neighboring reference SNPs in LD with it and their heterozygosities). We present two ways of constructing the model pdf for z-scores, using multinomial expansion, and using convolution. The former is perhaps more intuitive, but the latter is more numerically tractable, yielding an exact solution, and is used here to obtain all reported results. The problem then is finding the three model parameters that give a maximum likelihood best fit for the model's prediction of the distribution of z-scores to the actual distribution of z-scores. Because we are fitting three parameters typically using $\gtrsim 10^6$ data points, it is appropriate to incorporate some data reduction to facilitate the computations. To that end, we bin the data (z-scores) into a 10 × 10 grid of heterozygosity-by-total LD (having tested different grid sizes to ensure convergence of results). Also, when building the LD and heterozygosity structures of reference SNPs, we fine-grained the LD range ($0 \leq r^2 \leq 1$), again ensuring that bins were small enough that results were well converged. To fit the model to the data we bin the z-scores (within each heterozygosity/total LD window) and calculate the multinomial probability for having the actual distribution of z-scores (numbers of z-scores in the z-score bins) given the model pdf for the distribution of z-scores, and adjusting the model parameters using a multidimensional unconstrained nonlinear minimization (Nelder-Mead), so as to maximize the likelihood of the data, given the parameters.

A visual summary of the predicted and actual distribution of z-scores is obtained by making quantile-quantile plots showing, for a wide range of significance thresholds going well beyond genome-wide significance, the proportion (x-axis) of typed SNPs exceeding any given threshold (y-axis) in the range. It is important also to assess the quantile-quantile sub-plots for SNPs in the heterozygosity-by-total LD grid elements (see S1 Appendix).

With the pdf in hand, various quantities can be calculated: the number of causal SNPs; the expected genetic effect (denoted $\delta$ below, where $\delta^2$ is the non-centrality parameter of a Chi-squared distribution) at the current sample size for a typed SNP given the SNP's z-score and its full LD and heterozygosity structure; the estimated SNP heritability, $h^2_{SNP}$ (excluding contributions from rare reference SNPs, i.e., with MAF<0.2%); and the sample size required to explain any percentage of that with genome-wide significant SNPs. The model can easily be extended using a more complex distribution for the underlying $\beta$'s, with multiple-component mixtures for small and large effects, and incorporating selection pressure through both heterozygosity dependence on effect sizes and linkage disequilibrium dependence on the prior probability of a SNP's being causal—issues we will address in future work.

## The model: Probability distribution for z-scores

To establish notation, we consider a bi-allelic genetic variant, $i$, and let $\beta_i$ denote the effect size of allele substitution of that variant on a given quantitative trait. We assume a simple additive generative model (simple linear regression, ignoring covariates) relating genotype to phenotype [18, 21]. That is, assume a linear vector equation (no summation over repeated indices)

$$y = g_i \beta_i + e_i \tag{1}$$

for phenotype vector $y$ over $N$ samples (mean-centered and normalized to unit variance), mean-centered genotype vector $g_i$ for the $i^{th}$ of $n$ SNPs (vector over samples of the additively coded number of reference alleles for the $i^{th}$ variant), true fixed effect $\beta_i$ (regression coefficient) for the SNP, and residual vector $e_i$ containing the effects of all the other causal SNPs, the independent random environmental component, and random error. Variants with non-zero fixed effect $\beta_i$ are said to be "causal". For SNP $i$, the estimated simple linear regression coefficient is

$$\hat{\beta}_i = g_i^T y / (g_i^T g_i) = \text{cov}(g_i, y)/\text{var}(g_i), \tag{2}$$

where $T$ denotes transpose and $g_i^T g_i / N = \text{var}(g_i) = H_i$ is the SNP's heterozygosity (frequency of the heterozygous genotype): $H_i = 2p_i(1 - p_i)$ where $p_i$ is the frequency of either of the SNP's alleles.

Consistent with the work of others [11, 15], we assume the causal SNPs are distributed randomly throughout the genome (an assumption that can be relaxed when explicitly considering different SNP categories, but that in the main is consistent with the additive variation explained by a given part of the genome being proportional to the length of DNA [22]). In a Bayesian approach, we assume that the parameter $\beta$ for a SNP has a distribution (in that specific sense, this is similar to a random effects model), representing subjective information on $\beta$, not a distribution across tangible populations [23]. Specifically, we posit a normal distribution for $\beta$ with variance given by a constant, $\sigma_\beta^2$:

$$\beta \sim \mathcal{N}(0, \sigma_\beta^2). \tag{3}$$

This is also how the $\beta$ are distributed across the set of causal SNPs. Therefore, taking into account all SNPs (the remaining ones are all null by definition), this is equivalent to the two-component Gaussian mixture model we originally proposed [20]

$$\beta \sim \pi_1 \mathcal{N}(0, \sigma_\beta^2) + (1 - \pi_1)\mathcal{N}(0, 0) \tag{4}$$

where $\mathcal{N}(0, 0)$ is the Dirac delta function, so that considering all SNPs, the net variance is $\text{var}(\beta) = \pi_1 \sigma_\beta^2$. If there is no LD (and assuming no source of spurious inflation), the association z-score for a SNP with heterozygosity $H$ can be decomposed into a fixed effect $\delta$ and a

residual random environment and error term, $\epsilon \sim \mathcal{N}(0, 1)$, which is assumed to be independent of $\delta$ [18]:

$$z = \delta + \epsilon \tag{5}$$

with

$$\delta = \sqrt{NH}\beta \tag{6}$$

so that

$$\begin{aligned}
\mathrm{var}(z) &= \mathrm{var}(\delta) + \mathrm{var}(\epsilon) \\
&\equiv \sigma^2 + 1
\end{aligned} \tag{7}$$

where

$$\sigma^2 = \sigma_\beta^2 NH. \tag{8}$$

By construction, under null, i.e., when there is no genetic effect, $\delta = 0$, so that $var(\epsilon) = 1$.

If there is no source of variance distortion in the sample, but there is a source of bias in the summary statistics for a subset of markers (e.g., the sample is composed of two or more subpopulations with different allele frequencies for a subset of markers—pure population stratification in the sample [24]), the marginal distribution of an individual's genotype at any of those markers will be inflated. The squared z-score for such a marker will then follow a non-central Chi-square distribution (with one degree of freedom); the non-centrality parameter will contain the causal genetic effect, if any, but biased up or down (confounding or loss of power, depending on the relative sign of the genetic effect and the SNP-specific bias term). The effect of bias shifts, arising for example due to stratification, is nontrivial, and currently not explicitly in our model; it is usually accounted for using standard methods [25].

Variance distortion in the distribution of z-scores can arise from cryptic relatedness in the sample (drawn from a population mixture with at least one subpopulation with identical-by-descent marker alleles, but no population stratification) [19]. If $z_u$ denotes the uninflated z-scores, then the inflated z-scores are

$$z = \sigma_0 z_u, \tag{9}$$

where $\sigma_0 \geq 1$ characterizes the inflation. Thus, from Eq 7, in the presence of inflation in the form of variance distortion

$$\begin{aligned}
\mathrm{var}(z) &= \sigma_0^2(\sigma^2 + 1) \\
&\equiv \tilde{\sigma}^2 + \sigma_0^2 \\
&\equiv \tilde{\sigma}_\beta^2 NH + \sigma_0^2
\end{aligned} \tag{10}$$

where $\tilde{\sigma}_\beta^2 \equiv \sigma_0^2 \sigma_\beta^2$, so that $\mathrm{var}(\delta) = \tilde{\sigma}^2 \equiv \tilde{\sigma}_\beta^2 NH$ and $\epsilon \sim \mathcal{N}(0, \sigma_0^2)$. In the presence of variance distortion one is dealing with inflated random variables $\tilde{\beta} \sim \mathcal{N}(0, \tilde{\sigma}_\beta^2)$, but we will drop the tilde on the $\beta$'s in what follows.

Since variance distortion leads to scaled z-scores [19], then, allowing for this effect in some of the extremely large data sets, we can assess the ability of the model to detect this inflation by artificially inflating the z-scores (Eq 9), and checking that the inflated $\hat{\sigma}_0^2$ is estimated correctly while the other parameter estimates remain unchanged.

Implicit in Eq 8 is approximating the denominator, $1 - q^2$, of the $\chi^2$ statistic non-centrality parameter to be 1, where $q^2$ is the proportion of phenotypic variance explained by the causal

variant, i.e., $q \equiv \sqrt{H}\beta$. So a more correct $\delta$ is

$$\delta = \sqrt{N}q/\sqrt{1-q^2}. \tag{11}$$

Taylor expanding in $q$ and then taking the variance gives

$$\mathrm{var}(\delta) = \sigma_\beta^2 NH[1 + (15/4)\sigma_\beta^4 H^2 + O(\sigma_\beta^8 H^4)]. \tag{12}$$

The additional terms will be vanishingly small and so do not contribute in a distributional sense; (quasi-) Mendelian or outlier genetic effects represent an extreme scenario where the model is not expected to be accurate, but SNPs for such traits are by definition easily detectable. So Eq 8 remains valid for the polygenicity of complex traits.

Now consider the effects of LD on z-scores. The simple linear regression coefficient estimate for typed SNP $i$, $\hat{\beta}_i$, and hence the GWAS z-score, implicitly incorporates contributions due to LD with neighboring causal SNPs. (A typed SNP is a SNP with a z-score, imputed or otherwise; generally these will compose a smaller set than that available in reference panels like 1000 Genomes used here for calculating the LD structure of typed SNPs.) In Eq 1, $e_i = \sum_{j \neq i} g_j \beta_j + \varepsilon$, where $g_j$ is the genotype vector for SNP $j$, $\beta_j$ is its true regression coefficient, and $\varepsilon$ is the independent true environmental and error residual vector (over the $N$ samples). Thus, explicitly including all causal true $\beta$'s, Eq 2 becomes

$$\hat{\beta}_i = \frac{\sum_j g_i^T g_j \beta_j}{NH_i} + \frac{g_i^T \varepsilon}{g_i^T g_i} \tag{13}$$
$$\equiv \beta_i' + \varepsilon_i'$$

(the sum over $j$ now includes SNP $i$ itself). This is the simple linear regression expansion of the estimated regression coefficient for SNP $i$ in terms of the independent latent (true) causal effects and the latent environmental (plus error) component; $\beta_i'$ is the effective simple linear regression expression for the true genetic effect of SNP $i$, with contributions from neighboring causal SNPs mediated by LD. Note that $g_i^T g_j / N$ is simply $\mathrm{cov}(g_i, g_j)$, the covariance between genotypes for SNPs $i$ and $j$. Since correlation is covariance normalized by the variances, $\beta_i'$ in Eq 13 can be written as

$$\beta_i' = \sum_j \sqrt{\frac{H_j}{H_i}} r_{ij} \beta_j. \tag{14}$$

where $r_{ij}$ is the correlation between genotypes at reference SNP $j$ and typed SNP $i$. Then, from Eq 5, the z-score for the typed SNP's association with the phenotype is given by:

$$z_i = \sqrt{NH_i}\beta_i' + \epsilon_i$$
$$= \sqrt{N}\sum_j \sqrt{H_j} r_{ij} \beta_j + \epsilon_i. \tag{15}$$

We noted that in the absence of LD, the distribution of the residual in Eq 5 is assumed to be univariate normal. But in the presence of LD (Eq 15) there are induced correlations. Letting $\epsilon$ denote the vector of residuals (with element $\epsilon_i$ for SNP $i$, $i = 1 \ldots n$), and $\mathbf{M}$ denote the (sparse) $n \times n$ LD-$r^2$ matrix, then, ignoring inflation, $\epsilon \sim \mathcal{N}(\mathbf{0}, \mathbf{M})$ [26]. Since the genotypes of two unrelated individuals are marginally independent, this multivariate normal distribution for $\epsilon$ is contingent on the summary statistics for all SNPs being determined from the *same* set of individuals, which generally is overwhelmingly, if not in fact entirely, the case (in the extreme,

with an independent set of individuals for each SNP, $M$ would be reduced to the identity matrix). A limitation of the present work is that we do not consider this complexity. This may account for the relatively minor misfit in the simulation results for cases of high polygenicity—see below.

Thus, for example, if the SNP itself is not causal but is in LD with $k$ causal SNPs that all have heterozygosity $H$, and where its LD with each of these is the same, given by some value $r^2$ ($0 < r^2 \leq 1$), then $\tilde{\sigma}^2$ in Eq 10 will be given by

$$\tilde{\sigma}^2 = kr^2\tilde{\sigma}_\beta^2 NH. \tag{16}$$

For this idealized case, the marginal distribution, or pdf, of z-scores for a set of such associated SNPs is

$$f_1(z; N, \mathcal{H}, \sigma_\beta, \sigma_0) = \phi(z; 0, kr^2\tilde{\sigma}_\beta^2 NH + \sigma_0^2) \tag{17}$$

where $\phi(\cdot; \mu, \sigma^2)$ is the normal distribution with mean $\mu$ and variance $\sigma^2$, and $\mathcal{H}$ is shorthand for the LD and heterozygosity structure of such SNPs (in this case, denoting exactly $k$ causal SNPs with LD given by $r^2$ and heterozygosity given by $H$). If a proportion $\alpha$ of all typed SNPs are similarly associated with the phenotype while the remaining proportion are all null (not causal and not in LD with causal SNPs), then the marginal distribution for all SNP z-scores is the Gaussian mixture

$$f(z) = (1 - \alpha)\phi(z; 0, \sigma_0^2) + \alpha f_1(z), \tag{18}$$

dropping the parameters for convenience.

For real genotypes, however, the LD and heterozygosity structure is far more complicated, and of course the causal SNPs are generally numerous and unknown. Thus, more generally, for each typed SNP $\mathcal{H}$ will be a two-dimensional histogram over LD ($r^2$) and heterozygosity ($H$), each grid element giving the number of SNPs falling within the edges of that ($r^2$, $H$) bin. Alternatively, for each typed SNP it can be built as two one-dimensional histograms, one giving the LD structure (counts of neighboring SNPs in each LD $r^2$ bin), and the other giving, for each $r^2$ bin, the mean heterozygosity for those neighboring SNPs, which will be accurate for sufficiently fine binning—within a bin, the heterozygosities of the tagged referene SNPs wll be in a vary narrow range. We use the latter in what follows. We present two consistent ways of expressing the *a posteriori* pdf for z-scores, based on multinomial expansion and on convolution, that provide complementary views. The multinomial approach perhaps gives a more intuitive feel for the problem, but the convolution approach is considerably more tractable numerically and is used here to obtain all reporter results. All code used in the analyses, including simulations, is publicly available on GitHub [27].

## Model PDF: Multinomial expansion

As in our previous work, we incorporate the model parameter $\pi_1$ for the fraction of all SNPs that are causal [18]. Additionally, we calculate the actual LD and heterozygosity structure for each SNP. That is, for each SNP we build a histogram of the numbers of other SNPs in LD with it for $w$ equally-spaced $r^2$-windows between $r_{min}^2$ and 1 where $r_{min}^2 = 0.05$ (approximately the noise floor for correlation when LD is calculated from the 503 samples in 1000 Genomes), and record the mean heterozygosity for each bin; as noted above, we use $\mathcal{H}$ as shorthand to represent all this. We find that $w \simeq 20$ is sufficient for converged results. For any given SNP, the set of SNPs thus determined to be in LD with it constitute its LD block, with their number given by $n$ (LD with self is always 1, so $n$ is at least 1). The pdf for z-scores, given $N$, $\mathcal{H}$, and the three model parameters $\pi_1$, $\sigma_\beta$, $\sigma_0$, will then be given by the sum of Gaussians that are

generalizations of Eq 17 for different combinations of numbers of causal SNPs among the $w$ LD windows, each Gaussian scaled by the probability of the corresponding combination of causal SNPs among the LD windows, i.e., by the appropriate multinomial distribution term.

For $w$ $r^2$-windows, we must consider the possibilities where the typed SNP is in LD with all possible numbers of causal SNPs in each of these windows, or any combination thereof. There are thus $w + 1$ categories of SNPs: null SNPs (which $r^2$-windows they are in is irrelevant), and causal SNPs, where it does matter which $r^2$-windows they reside in. If window $i$ has $n_i$ SNPs ($\sum_{i=1}^{w} n_i = n$) and mean heterozygosity $H_i$, and the overall fraction of SNPs that are causal is $\pi_1$, then the probability of having simultaneously $k_0$ null SNPs, $k_1$ causal SNPs in window 1, and so on through $k_w$ causal SNPs in window $w$, for a nominal total of $K$ causal SNPs ($\sum_{i=1}^{w} k_i = K$ and $k_0 = n - K$), is given by the multinomial distribution, which we denote $M(k_0, \ldots, k_w; n_0, \ldots, n_w; \pi_1)$. For an LD block of $n$ SNPs, the prior probability, $p_i$, for a SNP to be causal and in window $i$ is the product of the independent prior probabilities of a SNP being causal and being in window $i$: $p_i = \pi_1 n_i/n$. The prior probability of being null (regardless of $r^2$-window) is simply $p_0 = (1 - \pi_1)$. The probability of a given breakdown $k_0, \ldots, k_w$ of the neighboring SNPs into the $w + 1$ categories is then given by

$$M(k_0, \ldots, k_w; n_0, \ldots, n_w; \pi_1) = \frac{n!}{k_0! \ldots k_w!} p_0^{k_0} \ldots p_w^{k_w} \tag{19}$$

and the corresponding Gaussian is

$$\phi(z; 0, (k_1 H_1 r_1^2 + \ldots + k_w H_w r_w^2)\tilde{\sigma}_\beta^2 N + \sigma_0^2). \tag{20}$$

For a SNP with LD and heterozygosity structure $\mathcal{H}$, the pdf for its z-score, given $N$ and the model parameters, is then given by summing over all possible numbers of total causal SNPs in LD with the SNP, and all possible distributions of those causal SNPs among the $w$ $r^2$-windows:

$$\begin{aligned} \text{pdf}(z; N, \mathcal{H}, \pi_1, \sigma_\beta, \sigma_0) = \\ \sum_{K=0}^{K_{max}} \sum_{k_1, \ldots, k_w} \frac{n!}{k_0! \ldots k_w!} p_0^{k_0} \ldots p_w^{k_w} \times \\ \phi(z; 0, (k_1 H_1 r_1^2 + \ldots + k_w H_w r_w^2)\tilde{\sigma}_\beta^2 N + \sigma_0^2), \end{aligned} \tag{21}$$

where $K_{max}$ is bounded above by $n$. Note again that $\mathcal{H}$ is shorthand for the heterozygosity and linkage-disequilibrium structure of the SNP, giving the set $\{n_i\}$ (as well as $\{H_i\}$), and hence, for a given $\pi_1$, $p_i$. Also there is the constraint $\sum_{i=1}^{w} k_i = K$ on the second summation, and, for all $i$, $\max(k_i) = \max(K, n_i)$, though generally $K_{max} \ll n_i$. The number of ways of dividing $K$ causal SNPs amongst $w$ LD windows is given by the binomial coefficient $\binom{a}{b}$, where $a \equiv K + w - 1$ and $b \equiv w - 1$, so the number of terms in the second summation grows rapidly with $K$ and $w$. However, because $\pi_1$ is small (often $\leq 10^{-3}$), the upper bound on the first summation over total number of potential causal SNPs $K$ in the LD block for the SNP can be limited to $K_{max} < \min(20, n)$, even for large blocks with $n \simeq 10^3$. That is,

$$\sum_{K=0}^{K_{max}} \sum_{k_1, \ldots, k_w} M(k_0, \ldots, k_w; n_0, \ldots, n_w; \pi_1) \simeq 1. \tag{22}$$

Still, the number of terms is large; e.g., for $K = 10$ and $w = 10$ there are 92,378 terms.

For any given typed SNP (whose z-score we are trying to predict), it is important to emphasize that the specific LD $r^2$ and the heterozygosity of each underlying causal (reference) SNP tagged by it need to be taken into account, at least in an approximate sense that can be controlled

to allow for arbitrary finessing giving converged results. This is the purpose of our $w = 20$ LD-$r^2$ windows, which inevitably leads to the multinomial expansion. Which window the causal SNP is in matters, leading to $w+1$ SNP categories, as noted above. Setting $w = 1$ would result in only a very rough approximation for the model pdf, reducing our multinomial to a binomial involving just two categories of SNPs: null and causal, with all causal SNPs treated the same, regardless of their LD with the tag SNP and their heterozygosity, as is done for the "M2" and "M3" models in [14]. The effects of this are demonstrated in the S1 Appendix (pp. S2-S3, and p. 14).

## Model PDF: Convolution

From Eq 15, there exists an efficient procedure that allows for accurate calculation of a z-score's *a posteriori* pdf (given the SNP's heterozygosity and LD structure, and the phenotype's model parameters). Any GWAS z-score is a sum of unobserved random variables (LD-mediated contributions from neighboring causal SNPs, and the additive environmental component), and the pdf for such a composite random variable is given by the convolution of the pdfs for the component random variables. Since convolution is associative, and the Fourier transform of the convolution of two functions is just the product of the individual Fourier transforms of the two functions, one can obtain the *a posteriori* pdf for z-scores as the inverse Fourier transform of the product of the Fourier transforms of the individual random variable components.

From Eq 15 $z$ is a sum of correlation- and heterozygosity-weighted random variables $\{\beta_j\}$ and the random variable $\epsilon$, where $\{\beta_j\}$ denotes the set of true causal parameters for each of the SNPs in LD with the typed SNP whose z-score is under consideration. The Fourier transform $F(k)$ of a Gaussian $f(x) = c \times exp(-ax^2)$ is $F(k) = c\sqrt{\pi/a} \times \exp(-\pi^2 k^2/a)$. From Eq 4, for each SNP j in LD with the typed SNP ($1 \leq j \leq b$, where $b$ is the typed SNP's block size),

$$\sqrt{NH_j}r_j\beta_j \sim \pi_1\mathcal{N}(0, NH_jr_j^2\tilde{\sigma}_\beta^2) + (1 - \pi_1)\mathcal{N}(0,0). \tag{23}$$

The Fourier transform (with variable $k$—see below) of the first term on the right hand side is

$$F(k) = \pi_1\exp(-2\pi^2 k^2 NH_jr_j^2\tilde{\sigma}_\beta^2), \tag{24}$$

while that of the second term is simply $(1 - \pi_1)$. Additionally, the environmental term is $\epsilon \sim \mathcal{N}(0, \sigma_0^2)$ (ignoring LD-induced correlation, as noted earlier), and its Fourier transform is $\exp(-2\pi^2\sigma_0^2 k^2)$. For each typed SNP, one could construct the *a posteriori* pdf based on these Fourier transforms. However, it is more practical to use a coarse-grained representation of the data. Thus, in order to fit the model to a data set, we bin the typed SNPs whose z-scores comprise the data set into a two-dimensional heterozygosity/total LD grid (whose elements we denote "H-L" bins), and fit the model with respect to this coarse grid instead of with respect to every individual typed SNP z-score; in the section "Parameter Estimation" below we describe using a $10 \times 10$ grid. Additionally, for each H-L bin the LD $r^2$ and heterozygosity histogram structure for each typed SNP is built, using $w_{max}$ equally-spaced $r^2$ bins for $r_{min}^2 \leq r^2 \leq 1$ (this is a change in notation from the previous section: $w_{max}$ here plays the role of $w$ there; in what follows, $w$ will be used as a running index, $1 \leq w \leq w_{max}$); $w_{max} = 20$ is large enough to allow for converged results; $r_{min}^2 = 0.05$ is generally small enough to capture true causal associations in weak LD while large enough to exclude spurious contributions to the pdf arising from estimates of $r^2$ that are non-zero due to noise. This points up a minor limitation of the model stemming from the small reference sample size ($N_R = 503$ for 1000 Genomes) from which $\mathcal{H}$ is built. Larger $N_R$ would allow for more precision in handling very low LD ($r^2 < 0.05$), but this

is an issue only for situations with extremely large $\sigma_\beta^2$ (high heritability with low polygenicity) that we do not encounter for the 16 phenotypes we analyze here. In any case, this can be calibrated for using simulations.

We emphasize again that setting $w_{max} = 1$ would result in only an approximation for the model pdf (see "Relation to Other Work" in the S1 Appendix).

For any H-L bin with mean heterozygosity $H$ and mean total LD $L$ there will be an average LD and heterozygosity structure with a mean breakdown for the typed SNPs having $n_w$ reference SNPs (not all of which necessarily are typed SNPs, i.e., have a z-score) with LD $r^2$ in the $w^{th}$ $r^2$ bin whose average heterozygosity is $H_w$. Thus, one can re-express z-scores for an H-L bin as

$$z = \sqrt{N} \sum_{w=1}^{w_{max}} \left( \sqrt{H_w} r_w \sum_{j=0}^{n_w} \beta_j \right) + \epsilon \qquad (25)$$

where $\beta_j$ and $\epsilon$ are unobserved random variables.

In the spirit of the discrete Fourier transform (DFT), discretize the set of possible z-scores into the ordered set of $n$ (equal to a power of 2) values $z_1, \ldots, z_n$ with equal spacing between neighbors given by $\Delta z$ ($z_n = -z_1 - \Delta z$, and $z_{n/2+1} = 0$). Taking $z_1 = -38$ allows for the minimum p-values of $5.8 \times 10^{-316}$ (near the numerical limit); with $n = 2^{10}$, $\Delta z = 0.0742$. Given $\Delta z$, the Nyquist critical frequency is $f_c = \frac{1}{2\Delta z}$, so we consider the Fourier transform function for the z-score pdf at $n$ discrete values $k_1, \ldots, k_n$, with equal spacing between neighbors given by $\Delta k$, where $k_1 = -f_c$ ($k_n = -k_1 - \Delta k$, and $k_{n/2+1} = 0$; the DFT pair $\Delta z$ and $\Delta k$ are related by $\Delta z \Delta k = 1/n$). Define

$$A_w \equiv -2\pi^2 N H_w r_w^2 \tilde{\sigma}_\beta^2. \qquad (26)$$

(see Eq 24). Then the product (over $r^2$ bins) of Fourier transforms for the genetic contribution to z-scores, denoted $G_j \equiv G(k_j)$, is

$$G(k_j) = \prod_{w=1}^{w_{max}} (\pi_1 \exp(A_w k_j^2) + (1 - \pi_1))^{n_w}. \qquad (27)$$

Recall that $\mathcal{H}$ denotes the LD and heterozygosity structure of a particular SNP (or representative SNP in an average sense for an H-L grid element), a shorthand for the set of values $\{n_w, H_w, L_w: w = 1, \ldots, w_{max}\}$ that characterize the SNP. Let $\mathcal{M}$ denote the set of model parameters. The Fourier transform of the environmental contribution, denoted $E_j \equiv E(k_j)$, is

$$E(k_j) = \exp(-2\pi^2 \sigma_0^2 k_j^2). \qquad (28)$$

Let $\mathbf{F_z} = (G_1 E_1, \ldots, G_n E_n)$ denote the vector of products of Fourier transform values, and let $\mathcal{F}^{-1}$ denote the inverse Fourier transform operator. Then for the SNP in question, the vector of pdf values, $\mathbf{pdf_z}$, for the uniformly discretized possible z-score outcomes $z_1, \ldots, z_n$ described above, i.e., $\mathbf{pdf_z} = (f_1, \ldots, f_n)$ where $f_i \equiv pdf(z_i | \mathcal{H}, \mathcal{M}, N)$, is

$$\mathbf{pdf_z} = \mathcal{F}^{-1} [\mathbf{F_z}]. \qquad (29)$$

Thus, the $i^{th}$ element $\mathbf{pdf}_{zi} = f_i$ is the *a posteriori* probability of obtaining a z-score value $z_i$ for the SNP, given the SNP's LD and heterozygosity structure, the model parameters, and the sample size.

## Data preparation

For real phenotypes, we calculated SNP minor allele frequency (MAF) and LD between SNPs using the 1000 Genomes phase 3 data set for 503 subjects/samples of European ancestry [28–30]. In order to carry out realistic simulations (i.e., with realistic heterozygosity and LD structures for SNPs), we used HAPGEN2 [31–33] to generate genotypes; we calculated SNP MAF and LD structure from 1000 simulated samples. We elected to use the same intersecting set of SNPs for real data and simulation. For HAPGEN2, we eliminated SNPs with MAF<0.002; for 1000 Genomes, we eliminated SNPs for which the call rate (percentage of samples with useful data) was less than 90%. This left $n_{snp}$ = 11,015,833 SNPs. See S1 Appendix for further details.

We analyzed summary statistics for sixteen phenotypes (S1 DataSourceList.; in what follows, where sample sizes varied by SNP, we quote the median value): (1) major depressive disorder ($N_{cases}$ = 59,851, $N_{controls}$ = 113,154) [34]; (2) bipolar disorder ($N_{cases}$ = 20,352, $N_{controls}$ = 31,358) [35]; (3) schizophrenia ($N_{cases}$ = 35,476, $N_{controls}$ = 46,839) [36]; (4) coronary artery disease ($N_{cases}$ = 60,801, $N_{controls}$ = 123,504) [37]; (5) ulcerative colitis ($N_{cases}$ = 12,366, $N_{controls}$ = 34,915) and (6) Crohn's disease ($N_{cases}$ = 12,194, $N_{controls}$ = 34,915) [38]; (7) late onset Alzheimer's disease (LOAD; $N_{cases}$ = 17,008, $N_{controls}$ = 37,154) [39] (in the S1 Appendix we present results for a more recent GWAS with $N_{cases}$ = 71,880 and $N_{controls}$ = 383,378 [40]); (8) amyotrophic lateral sclerosis (ALS) ($N_{cases}$ = 12,577, $N_{controls}$ = 23,475) [41]; (9) number of years of formal education ($N$ = 293,723) [42]; (10) intelligence ($N$ = 262,529) [43, 44]; (11) body mass index ($N$ = 233,554) [45]; (12) height ($N$ = 251,747) [46]; (13) putamen volume (normalized by intracranial volume, $N$ = 11,598) [47]; (14) low- ($N$ = 89,873) and (15) high-density lipoprotein ($N$ = 94,295) [48]; and (16) total cholesterol ($N$ = 94,579) [48]. Most participants were of European ancestry.

For height, we focused on the 2014 GWAS [46], not the more recent 2018 GWAS [49], although we also report below model results for the latter. There are issues pertaining to population structure in the various height GWAS [50, 51], and the 2018 GWAS is a combination of GIANT and UKB GWAS, so some caution is warranted in interpreting results for these data.

For the ALS GWAS data, there is very little signal outside chromosome 9: the data QQ plot essentially tracks the null distribution straight line. The QQ plot for chromosome 9, however, shows a significant departure from the null distribution. Of 471,607 SNPs on chromosome 9 a subset of 273,715 have z-scores, of which 107 are genome-wide significant, compared with 114 across the full genome. Therefore, we restrict ALS analysis to chromosome 9.

A limitation in the current work is that we have not taken account of imputation inaccuracy, where lower MAF SNPs are, through lower LD, less certain. Thus, the effects from lower MAF causal variants will be noisier than for higher MAF variants.

## Simulations

We generated genotypes for $10^5$ unrelated simulated samples using HAPGEN2 [33]. For narrow-sense heritability $h^2$ equal to 0.1, 0.4, and 0.7, we considered polygenicity $\pi_1$ equal to $10^{-5}$, $10^{-4}$, $10^{-3}$, and $10^{-2}$. For each of these 12 combinations, we randomly selected $n_{causal} = \pi_1 \times n_{snp}$ "causal" SNPs and assigned them $\beta$-values drawn from the standard normal distribution (i.e., independent of $H$), with all other SNPs having $\beta = 0$. We repeated this ten times, giving ten independent instantiations of random vectors of $\beta$'s. Defining $Y_G = G\beta$, where $G$ is the genotype matrix and $\beta$ here is the vector of true coefficients over all SNPs, the total phenotype vector is constructed as $Y = Y_G + \varepsilon$, where the residual random vector $\varepsilon$ for each instantiation is drawn from a normal distribution such that $h^2 = \text{var}(Y_G)/\text{var}(Y)$. For each of the instantiations this implicitly defines the "true" value $\sigma_\beta^2$.

The sample simple linear regression slope, $\hat{\beta}$, and the Pearson correlation coefficient, $\hat{r}$, are assumed to be t-distributed. These quantities have the same t-value:

$t = \hat{\beta}/\text{se}(\hat{\beta}) = \hat{r}/\text{se}(\hat{r}) = \hat{r}\sqrt{N-2}/\sqrt{1-\hat{r}^2}$, with corresponding p-value from Student's $t$ cumulative distribution function (cdf) with $N-2$ degrees of freedom: $p = 2 \times \text{tcdf}(-|t|, N-2)$ (see S1 Appendix). Since we are not here dealing with covariates, we calculated $p$ from correlation, which is slightly faster than from estimating the regression coefficient. The t-value can be transformed to a z-value, giving the z-score for this $p$: $z = -\Phi^{-1}(p/2) \times \text{sign}(\hat{r})$, where $\Phi$ is the normal cdf ($z$ and $t$ have the same p-value).

## Parameter estimation

We randomly pruned SNPs using the threshold $r^2 > 0.8$ to identify "synonymous" SNPs, performing ten such iterations. That is, for each of ten iterations, we randomly selected a SNP (not necessarily the one with largest z-score) to represent each subset of synonymous SNPs. For schizophrenia, for example, pruning resulted in approximately 1.3 million SNPs in each iteration.

The postulated pdf for a SNP's z-score depends on the SNP's LD and heterozygosity structure (histogram), $\mathcal{H}$. Given the data–the set of z-scores for available SNPs, as well as their LD and heterozygosity structure—and the $\mathcal{H}$-dependent pdf for z-scores, the objective is to find the model parameters that best predict the distribution of z-scores. We bin the SNPs with respect to a grid of heterozygosity and total LD; for any given H-L bin there will be a range of z-scores whose distribution the model it intended to predict. We find that a $10 \times 10$ grid of equally spaced bins is adequate for converged results. (Using equally-spaced bins might seem inefficient because of the resulting very uneven distribution of z-scores among grid elements—for example, orders of magnitude more SNPs in grid elements with low total LD compared with high total LD. However, the objective is to model the effects of H and L: using variable grid element sizes so as to maximize balance of SNP counts among grid elements means that the true H- and L-mediated effects of the SNPs in a narrow range of H and L get subsumed with the effects of many more SNPs in a much wider range of H and L—a misspecification of the pdf leading to some inaccuracy.) In lieu of or in addition to total LD (L) binning, one can bin SNPs with respect to their total LD block size (total number of SNPs in LD, ranging from 1 to $\simeq 1,500$).

To find the model parameters that best fit the data, for a given H-L bin we binned the selected SNPs z-scores into equally-spaced bins of width $dz = 0.0742$ (between $z_{min} = -38$ and $z_{max} = 38$, allowing for p-values near the numerical limit of $10^{-316}$), and from Eq 29 calculated the probability for z-scores to be in each of those z-score bins (the prior probability for "success" in each z-score bin). Then, knowing the actual numbers of z-scores (numbers of "successes") in each z-score bin, we calculated the multinomial probability, $p_m$, for this outcome. The optimal model parameter values will be those that maximize the accrual of this probability over all H-L bins. We constructed a cost function by calculating, for a given H-L bin, $-\ln(p_m)$ and averaging over prunings, and then accumulating this over all H-L bins. Model parameters minimizing the cost were obtained from Nelder-Mead multidimensional unconstrained nonlinear minimization of the cost function, using the Matlab function `fminsearch()`.

## Posterior effect sizes

Model posterior effect sizes, given $z$ (along with N, $\mathcal{H}$, and the model parameters), were calculated using numerical integration over the random variable $\delta$:

$$
\begin{aligned}
\delta_{expected} \equiv E(\delta|z) \quad &= \int P(\delta|z)\delta d\delta \\
&= \frac{1}{P(z)} \int P(z|\delta)P(\delta)\delta d\delta.
\end{aligned}
\tag{30}
$$

Here, since $z|\delta \sim \mathcal{N}(\delta, \sigma_0^2)$, the posterior probability of $z$ given $\delta$ is simply

$$P(z|\delta) = \phi(z; \delta, \sigma_0^2). \tag{31}$$

$P(z)$ is shorthand for $\mathrm{pdf}(z|N, \mathcal{H}, \pi_1, \sigma_\beta, \sigma_0)$, given by Eq 29. $P(\delta)$ is calculated by a similar procedure that lead to Eq 29 but ignoring the environmental contributions $\{E_j\}$. Specifically, let $\mathbf{F}_\delta = (G_1, \ldots, G_n)$ denote the vector of products of Fourier transform values. Then, the vector of pdf values for genetic effect bins (indexed by $i$; numerically, these will be the same as the z-score bins) in the H-L bin, $\mathbf{pdf}_\delta = (f_1, \ldots, f_n)$ where $f_i \equiv \mathrm{pdf}(\delta_i|\mathcal{H})$, is

$$\mathbf{pdf}_\delta = \mathcal{F}^{-1}[\mathbf{F}_\delta]. \tag{32}$$

Similarly,

$$\begin{aligned} \delta_{expected}^2 \equiv E(\delta^2|z) &= \int P(\delta|z)\delta^2 d\delta \\ &= \frac{1}{P(z)} \int P(z|\delta)P(\delta)\delta^2 d\delta, \end{aligned} \tag{33}$$

which is used in power calculations.

## GWAS replication

A related matter has to do with whether z-scores for SNPs reaching genome-wide significance in a discovery-sample are compatible with the SNPs' z-scores in a replication-sample, particularly if any of those replication-sample z-scores are far from reaching genome-wide significance, or whether any apparent mismatch signifies some overlooked inconsistency. The model pdf allows one to make a principled statistical assessment in such cases. We present the details for this application, and results applied to studies of bipolar disorder, in the S1 Appendix (pp. S7-S11).

## GWAS power

Chip heritability, $h_{SNP}^2$, is the proportion of phenotypic variance that in principle can be captured additively by the $n_{snp}$ SNPs under study [17]. It is of interest to estimate the proportion of $h_{SNP}^2$ that can be explained by SNPs reaching genome-wide significance, $p \leq 5 \times 10^{-8}$ (i.e., for which $|z| > z_t = 5.45$), at a given sample size [52, 53]. In Eq 1, for SNP $i$ with genotype vector $g_i$ over $N$ samples, let $y_{g_i} \equiv g_i\beta_i$. If the SNP's heterozygosity is $H_i$, then $\mathrm{var}(y_{g_i}) = \beta_i^2 H_i$. If we knew the full set $\{\beta_i\}$ of true $\beta$-values, then, for z-scores from a particular sample size $N$, the proportion of SNP heritability captured by genome-wide significant SNPs, $A(N)$, would be given by

$$A(N) = \frac{\sum_{i:|z_i|>z_t} \beta_i^2 H_i}{\sum_{all\ i} \beta_i^2 H_i}. \tag{34}$$

Now, from Eq 15, $\delta_i = \sqrt{N} \sum_j \sqrt{H_j} r_{ij} \beta_j$. If SNP $i$ is causal and sufficiently isolated so that it is not in LD with other causal SNPs, then $\delta_i = \sqrt{N}\sqrt{H_i}\beta_i$, and $\mathrm{var}(y_{g_i}) = \delta_i^2/N$. When all causal SNPs are similarly isolated, Eq 34 becomes

$$A(N) = \frac{\sum_{i:|z_i|>z_t} \delta_i^2}{\sum_{all\ i} \delta_i^2}. \tag{35}$$

Of course, the true $\beta_i$ are not known and some causal SNPs will likely be in LD with others. Furthermore, due to LD with causal SNPs, many SNPs will have a nonzero (latent or unobserved) effect size, $\delta$. Nevertheless, we can formulate an approximation to $A(N)$ which, assuming the pdf for z-scores (Eq 29) is reasonable, will be inaccurate to the degree that the average LD structure of genome-wide significant SNPs differs from the overall average LD structure. As before (see the subsection "Model PDF: Convolution"), consider a fixed set of $n$ equally-spaced nominal z-scores covering a wide range of possible values (changing from the summations in Eq 35 to the uniform summation spacing $\Delta z$ now requires bringing the probability density into the summations). For each $z$ from the fixed set (and, as before, employing data reduction by averaging so that H and L denote values for the $10 \times 10$ grid), use $E(\delta^2|z, N, H, L)$ given in Eq 33 to define

$$C(z|N, H, L) \equiv E(\delta^2|z, N, H, L)P(z|N, H, L) \tag{36}$$

(emphasizing dependence on N, H, and L). Then, for any N, $A(N)$ can be estimated by

$$A(N) = \frac{\sum_{H,L} \sum_{z:|z|>z_t} C(z, N, H, L)}{\sum_{H,L} \sum_{all\ z} C(z, N, H, L)} \tag{37}$$

where $\Sigma_{H,L}$ denotes sum over the H-L grid elements. The ratio in Eq 37 should be accurate if the average effects of LD in the numerator and denominator cancel—which will always be true as the ratio approaches 1 for large N. Plotting $A(N)$ gives an indication of the power of future GWAS to capture chip heritability.

## Quantile-quantile plots and genomic control

One of the advantages of quantile-quantile (QQ) plots is that on a logarithmic scale they emphasize behavior in the tails of a distribution, and provide a valuable visual aid in assessing the independent effects of polygenicity, strength of association, and variance distortion—the roles played by the three model parameters–as well as showing how well a model fits data. QQ plots for the model were constructed using Eq 29, replacing the normal pdf with the normal cdf, and replacing $z$ with an equally-spaced vector $\vec{z}_{nom}$ of length 10,000 covering a wide range of nominal $|z|$ values (0 through 38). SNPs were divided into a $10 \times 10$ grid of H × L bins, and the cdf vector (with elements corresponding to the z-values in $\vec{z}_{nom}$) accumulated for each such bin (using mean values of H and L for SNPs in a given bin).

For a given set of samples and SNPs, the genomic control factor, $\lambda$, for the z-scores is defined as the median $z^2$ divided by the median for the null distribution, 0.455 [19]. This can also be calculated from the QQ plot. In the plots we present here, the abscissa gives the -$\log_{10}$ of the proportion, $q$, of SNPs whose z-scores exceed the two-tailed significance threshold $p$, transformed in the ordinate as -$\log_{10(p)}$. The median is at $q_{med} = 0.5$, or $-\log_{10}(q_{med}) \simeq 0.3$; the corresponding empirical and model p-value thresholds ($p_{med}$) for the z-scores—and equivalently for the z-scores-squared—can be read off from the plots. The genomic inflation factor is then given by

$$\lambda = [\Phi^{-1}(p_{med}/2)]^2/0.455.$$

Note that the values of $\lambda$ reported here are for pruned SNP sets; these values will be lower than for the total GWAS SNP sets.

Knowing the total number, $n_{tot}$, of p-values involved in a QQ plot (number of GWAS z-scores from pruned SNPs), any point $(q, p)$ (log-transformed) on the plot gives the number, $n_p = qn_{tot}$, of p-values that are as extreme as or more extreme than the chosen p-value. This

can be thought of as $n_p$ "successes" out of $n_{tot}$ independent trials (thus ignoring LD) from a binomial distribution with prior probability $q$. To approximate the effects of LD, we estimate the number of independent SNPs as $n_{tot}/f$ where $f \simeq 10$. The 95% binomial confidence interval for $q$ is calculated as the exact Clopper-Pearson 95% interval [54], which is similar to the normal approximation interval, $q \pm 1.96\sqrt{q(1-q)/n_{tot}/f}$.

## Number of causal SNPs

The estimated number of causal SNPs is given by the polygenicity, $\pi_1$, times the total number of SNPs, $n_{snp}$: $n_{causal} = \pi_1 n_{snp}$. $n_{snp}$ is given by the total number of SNPs that went into building the heterozygosity/LD structure, $\mathcal{H}$ in Eq 29, i.e., the approximately 11 million SNPs selected from the 1000 Genomes Phase 3 reference panel, not the number of typed SNPs in the particular GWAS. The parameters estimated are to be seen in the context of the reference panel, which we assume contains all common causal variants. Stable quantities (i.e., fairly independent of the reference panel size. e.g., using the full panel or ignoring every second SNP), are the estimated effect size variance and number of causal variants—which we demonstrate below—and hence the heritability. Thus, the polygenicity will scale inversely with the reference panel size. A reference panel with a substantially larger number of samples would allow for inclusion of more SNPs (non-zero MAF), and thus the actual polygenicity estimated would change slightly.

## Narrow-sense chip heritability

Since we are treating the $\beta$ coefficients as fixed effects in the simple linear regression GWAS formalism, with the phenotype vector standardized with mean zero and unit variance, from Eq 1 the proportion of phenotypic variance explained by a particular causal SNP whose reference panel genotype vector is $g$, $q^2 = \text{var}(y;g)$, is given by $q^2 = \beta^2 H$. The proportion of phenotypic variance explained additively by all causal SNPs is, by definition, the narrow sense chip heritability, $h^2$. Since $E(\beta^2) = \sigma_\beta^2$ and $n_{causal} = \pi_1 n_{snp}$, and taking the mean heterozygosity over causal SNPs to be approximately equal to the mean over all SNPs, $\bar{H}$, the chip heritability can be estimated as

$$h^2 = \pi_1 n_{snp} \bar{H} \sigma_\beta^2. \tag{38}$$

Mean heterozygosity from the $\simeq 11$ million SNPs is $\bar{H} = 0.2165$.

For all-or-none traits like disease status, the estimated $h^2$ from Eq 38 for an ascertained case-control study is on the observed scale and is a function of the prevalence in the adult population, $K$, and the proportion of cases in the study, $P$. The heritability on the underlying continuous liability scale [55], $h_l^2$, is obtained by adjusting for ascertainment (multiplying by $K(1 - K)/(P(1 - P))$, the ratio of phenotypic variances in the population and in the study) and rescaling based on prevalence [6, 56]:

$$h_l^2 = h^2 \frac{K(1-K)}{P(1-P)} \times \frac{K(1-K)}{a^2}, \tag{39}$$

where $a$ is the height of the standard normal pdf at the truncation point $z_K$ defined such that the area under the curve in the region to the right of $z_K$ is $K$.

## Confidence intervals

Confidence intervals for parameters were estimated using the inverse of the observed Fisher information matrix (FIM). The full FIM was estimated for all three parameters used in the

model. For the derived quantity $h^2$, which depends on all parameters, the covariances among the parameters, given by the off-diagonal elements of the inverse of the FIM, were incorporated. Numerical values are in S1 Appendix (p. S12).

## Results

### Simulations

Table 1 shows the simulation results, comparing true and estimated values for the model parameters, heritability, and the number of causal SNPs, for twelve scenarios where $\pi_1$ and $\sigma_\beta^2$ both range over three orders of magnitude, encompassing the range of values for the phenotypes; in S1 Appendix (p. S15) are QQ plots for a randomly chosen (out of 10) $\beta$-vector and phenotype instantiation for each of the twelve $(\pi_1, h^2)$ scenarios. Most of the $\hat{\pi}_1$ estimates are in very good agreement with the true values, though for the extreme scenario of high heritability and low polygenicity it is overestimated by factors of two-to-three. The numbers of estimated causal SNPs (out of $\simeq$11 million) are in correspondingly good agreement with the true values, ranging in increasing powers of 10 from 110 through 110,158. The estimated discoverabilities ($\hat{\sigma}_\beta^2$) are also in good agreement with the true values. In most cases, $\hat{\sigma}_0^2$ is close to 1, indicating little or no global inflation, though it is elevated for high heritability with high polygenicity, suggesting it is capturing some ubiquitous effects.

In the S1 Appendix (pp. S5-S7) we examine the issue of model misspecification. Specifically, we assign causal effects $\beta$ drawn from a Gaussian whose variance is not simply a constant but depends on heterozygosity, such that rarer causal SNPs will tend to have larger effects [15]. The results—see S1 Appendix (p. S5)–show that the model still makes reasonable estimates of the underlying genetic architecture. Additionally, we tested the scenario where true causal effects are distributed with respect to two Gaussians [14], a situation that allows for a small number of the causal SNPs to have quite large effects—see S1 Appendix (p. S6). We find that heritabilities are still reasonably estimated using our model. In all these scenarios the overall data QQ plots were accurately reproduced by the model. As a counter example, we simulated summary statistics where the prior probability of a reference SNP being causal decreased linearly with total LD (see S1 Appendix (p. S7)). In this case, our single Gaussian fit (which

**Table 1. Simulation results: Comparison of mean (std) true and estimated (^) model parameters and derived quantities.** Results for each line, for specified heritability $h^2$ and fraction $\pi_1$ of causal SNPs, are from 10 independent instantiations with random selection of the $n_{causal}$ causal SNPs that are assigned a $\beta$-value from the standard normal distribution. Defining $Y_g = G\beta$, where $G$ is the genotype matrix, the total phenotype vector is constructed as $Y = Y_g + \varepsilon$, where the residual random vector $\varepsilon$ for each instantiation is drawn from a normal distribution such that var$(Y)$ = var$(Y_g)/h^2$ for predefined $h^2$. For each of the instantiations, $i$, this implicitly defines the true value $\sigma_{\beta i}^2$, and $\sigma_\beta^2$ is their mean. An example QQ plot for each line entry is shown in in S1 Appendix (p. S15).

| $h^2$ | $\hat{h}^2$ | | $\pi_1$ | $\hat{\pi}_1$ | | $\sigma_\beta^2$ | | $\hat{\sigma}_\beta^2$ | | $\hat{\sigma}_0^2$ | | $n_{causal}$ | $\hat{n}_{causal}$ | |
|---|---|---|---|---|---|---|---|---|---|---|---|---|---|---|
| 0.1 | 0.12 | (0.01) | 1E-5 | 1.4E-5 | (2E-6) | 4.3E-3 | (7E-4) | 3.6E-3 | (5E-4) | 1.01 | (0.002) | 110 | 151 | (20) |
| 0.1 | 0.10 | (0.01) | 1E-4 | 1.0E-4 | (2E-5) | 4.2E-4 | (2E-5) | 4.1E-4 | (5E-5) | 1.01 | (0.003) | 1101 | 1130 | (206) |
| 0.1 | 0.09 | (0.01) | 1E-3 | 0.9E-3 | (1E-4) | 4.2E-5 | (5E-7) | 4.1E-5 | (4E-6) | 1.02 | (0.003) | 11015 | 10340 | (1484) |
| 0.1 | 0.09 | (0.01) | 1E-2 | 0.8E-2 | (2E-3) | 4.2E-6 | (4E-8) | 5.6E-6 | (2E-6) | 1.02 | (0.002) | 110158 | 83411 | (25448) |
| 0.4 | 0.52 | (0.05) | 1E-5 | 2.3E-5 | (2E-6) | 1.7E-2 | (3E-3) | 9.1E-3 | (1E-3) | 1.02 | (0.002) | 110 | 259 | (20) |
| 0.4 | 0.45 | (0.02) | 1E-4 | 1.2E-4 | (8E-6) | 1.7E-3 | (7E-5) | 1.5E-3 | (9E-5) | 1.04 | (0.002) | 1101 | 1310 | (92) |
| 0.4 | 0.39 | (0.01) | 1E-3 | 1.0E-3 | (5E-5) | 1.7E-4 | (2E-6) | 1.6E-4 | (8E-6) | 1.05 | (0.003) | 11015 | 10607 | (578) |
| 0.4 | 0.37 | (0.01) | 1E-2 | 0.9E-2 | (1E-3) | 1.7E-5 | (2E-7) | 1.7E-5 | (2E-6) | 1.06 | (0.003) | 110158 | 95135 | (10851) |
| 0.7 | 0.91 | (0.09) | 1E-5 | 2.9E-5 | (2E-6) | 3.0E-2 | (5E-3) | 1.3E-2 | (2E-3) | 1.02 | (0.003) | 110 | 324 | (24) |
| 0.7 | 0.82 | (0.02) | 1E-4 | 1.4E-4 | (7E-6) | 2.9E-3 | (1E-4) | 2.4E-3 | (1E-4) | 1.05 | (0.002) | 1101 | 1493 | (79) |
| 0.7 | 0.70 | (0.01) | 1E-3 | 1.0E-3 | (4E-5) | 2.9E-4 | (4E-6) | 2.8E-4 | (1E-5) | 1.08 | (0.003) | 11015 | 10866 | (406) |
| 0.7 | 0.66 | (0.01) | 1E-2 | 0.9E-2 | (7E-4) | 2.9E-5 | (3E-7) | 2.9E-5 | (2E-6) | 1.09 | (0.003) | 110158 | 95067 | (8191) |

assumes no LD dependence on the prior probability of a reference SNP being causal) did not produce model QQ plots that accurately tracked the data QQ plots (see S1 Appendix (p. S19)). The model parameters and heritabilities were also poor. But this scenario is highly artificial; in contrast, in situations where the data QQ plots were accurately reproduced by the model, the estimated model parameters and heritability were plausible.

## Phenotypes

Figs 1 and 2 show QQ plots for the pruned z-scores for eight qualitative and eight quantitative phenotypes, along with model estimates (S1 Appendix (pp. S20-S38) show a $4 \times 4$ grid breakdown with respect to heterozygosity × total-LD of QQ plots for all phenotypes studied here; the $4 \times 4$ grid is a subset of the $10 \times 10$ grid used in the calculations). In all cases, the model fit (yellow) closely tracks the data (dark blue). For the sixteen phenotypes, estimates for the model polygenicity parameter (fraction of reference panel, with $\simeq 11$ million SNPs, estimated to have non-null effects) range over two orders of magnitude, from $\pi_1 \simeq 2 \times 10^{-5}$ to $\pi_1 \simeq 4 \times 10^{-3}$. The estimated SNP discoverability parameter (variance of $\beta$, or expected $\beta^2$, for causal variants) also ranges over two orders of magnitude from $\sigma_\beta^2 \simeq 7 \times 10^{-6}$ to $\sigma_\beta^2 \simeq 2 \times 10^{-3}$ (in units where the variance of the phenotype is normalized to 1).

We find that schizophrenia and bipolar disorder appear to be similarly highly polygenic, with model polygenicities $\simeq 2.84 \times 10^{-3}$ and $\simeq 2.70 \times 10^{-3}$, respectively. The model

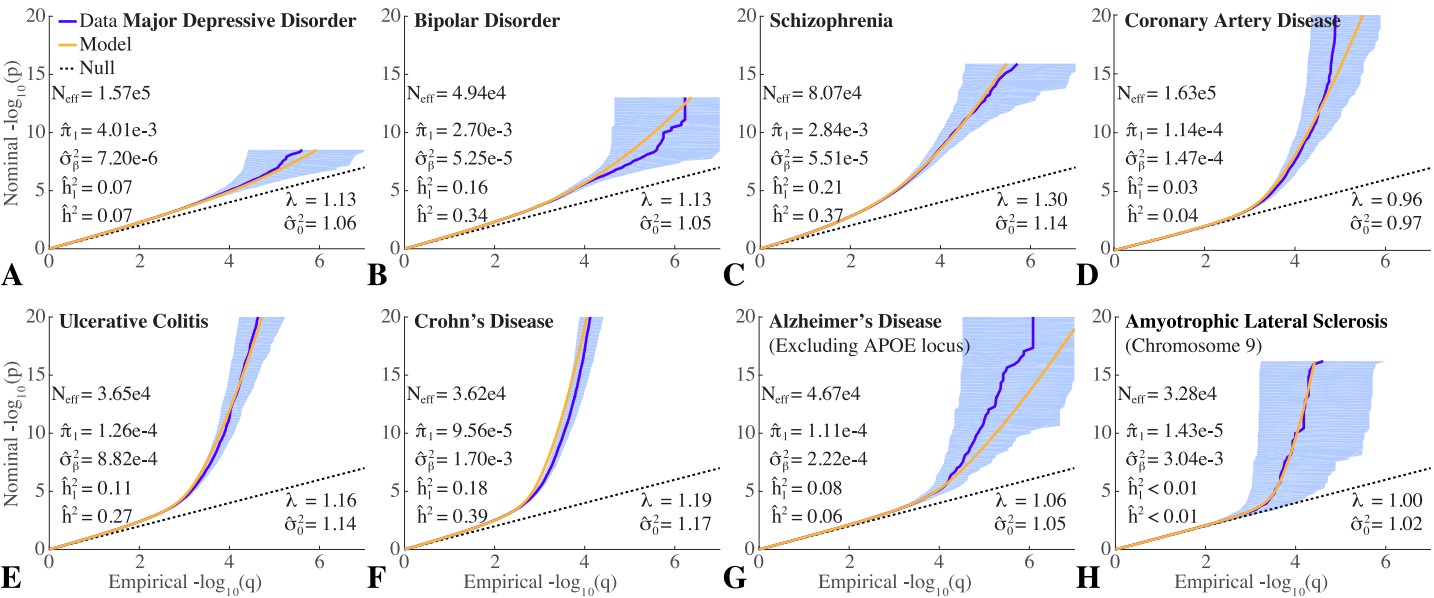

**Fig 1. QQ plots of (pruned) z-scores for qualitative phenotypes (dark blue, 95% confidence interval in light blue) with model prediction (yellow): (A) major depressive disorder; (B) bipolar disorder; (C) schizophrenia; (D) coronary artery disease (CAD); (E) ulcerative colitis (UC); (F) Crohn's disease (CD); (G) late onset Alzheimer's disease (AD), excluding APOE (see also S1 Appendix (p. S17)); and (H) amyotrophic lateral sclerosis (ALS), restricted to chromosome 9 (see also S1 Appendix (p. S18)).** The dashed line is the expected QQ plot under null (no SNPs associated with the phenotype). $p$ is a nominal p-value for z-scores, and $q$ is the proportion of z-scores with p-values exceeding that threshold. $\lambda$ is the overall nominal genomic control factor for the pruned data (which is accurately predicted by the model in all cases). The three estimated model parameters are: polygenicity, $\hat{\pi}_1$; discoverability, $\hat{\sigma}_\beta^2$ (corrected for inflation); and SNP association $\chi^2$-statistic inflation factor, $\hat{\sigma}_0^2$. $\hat{h}^2$ is the estimated narrow-sense chip heritability, re-expressed as $h_l^2$ on the liability scale for these case-control conditions assuming a prevalence of: MDD 7.1% [57], BIP 0.5% [58], SCZ 1% [59], CAD 3% [60], UC 0.1% [61], CD 0.1% [61], AD 14% (for people aged 71 and older in the USA [62, 63]), and ALS $5 \times 10^{-5}$ [64]. The estimated number of causal SNPs is given by $\hat{n}_{causal} = \hat{\pi}_1 n_{snp}$ where $n_{snp} = 11,015,833$ is the total number of SNPs, whose LD structure and MAF underlie the model; the GWAS z-scores are for subsets of these SNPs. $N_{eff}$ is the effective case-control sample size–see text. Reading the plots: on the vertical axis, choose a p-value threshold (more extreme values are further from the origin), then the horizontal axis gives the proportion of SNPs exceeding that threshold (higher proportions are closer to the origin). Numerical values for the model parameters are also given in Table 2. See also S1 Appendix (pp. S20-S28).

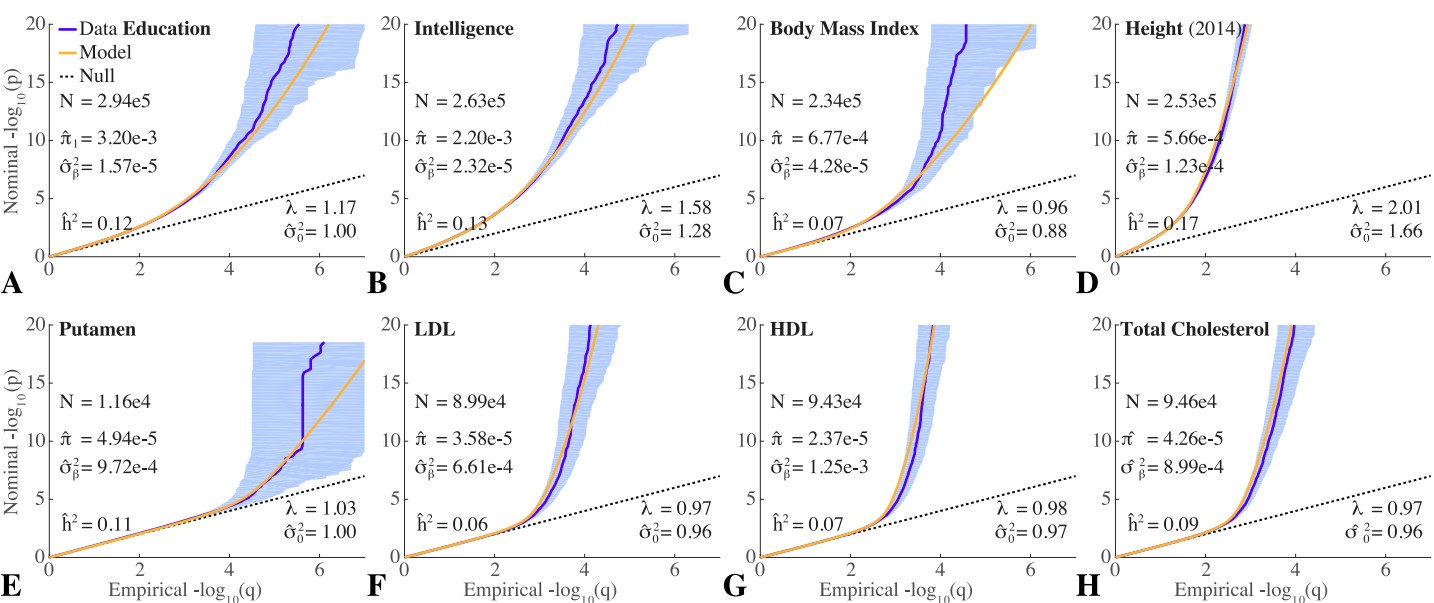

**Fig 2. this area for notes QQ plots of (pruned) z-scores and model fits for quantitative phenotypes:** (A) educational attainment; (B) intelligence; (C) body mass index (BMI); (D) height; (E) putamen volume; (F) low-density lipoprotein (LDL); (G) high-density lipoprotein (HDL); and (H) total cholesterol (TC). *N* is the sample size. See Fig 1 for further description. Numerical values for the model parameters are also given in Table 2. See also S1 Appendix (pp. S29-S38).

polygenicity of major depressive disorder, however, is 40% higher, $\pi_1 \simeq 4 \times 10^{-3}$—the highest value among the sixteen phenotypes. In contrast, the model polygenicities of late onset Alzheimer's disease and Crohn's disease are almost thirty times smaller than that of schizophrenia.

In S1 Appendix (p. S17) we show results for Alzheimer's disease exclusively for chromosome 19 (which contains APOE), and for all autosomal chromosomes excluding chromosome 19. We also show results with the same chromosomal breakdown for a recent GWAS involving 455,258 samples that included 24,087 clinically diagnosed LOAD cases and 47,793 AD-by-proxy cases (individuals who were not clinically diagnosed with LOAD but for whom at least one parent had LOAD) [65]. These GWAS give consistent estimates of polygenicity: $\pi_1 \sim 1 \times 10^{-4}$ excluding chromosome 19, and $\pi_1 \sim 6 \times 10^{-5}$ for chromosome 19 exclusively.

Of the quantitative traits, educational attainment has the highest model polygenicity, $\pi_1 = 3.2 \times 10^{-3}$, similar to intelligence, $\pi_1 = 2.2 \times 10^{-3}$. Approximately two orders of magnitude lower in polygenicity are the endophenotypes putamen volume and low- and high-density lipoproteins.

The model effective SNP discoverability for schizophrenia is $\hat{\sigma}_\beta^2 = 5.51 \times 10^{-5}$, similar to that for bipolar disorder. Major depressive disorder, which has the highest polygenicity, has the lowest SNP discoverability, approximately one-eighth that of schizophrenia; it is this low value, combined with high polygenicity that leads to the weak signal in Fig 1 (A) even though the sample size is relatively large. In contrast, SNP discoverability for Alzheimer's disease is almost four times that of schizophrenia. The inflammatory bowel diseases, however, have much higher SNP discoverabilities, 16 and 31 times that of schizophrenia respectively for ulcerative colitis and Crohn's disease—the latter having the second highest value of the sixteen phenotypes: $\hat{\sigma}_\beta^2 = 1.7 \times 10^{-3}$.

Additionally, for Alzheimer's disease we show in the S1 Appendix (p. S17) that the discoverability is two orders of magnitude greater for chromosome 19 than for the remainder of the autosome. Note that since two-thirds of the 2018 "cases" are AD-by-proxy, the discoverabilities

for the 2018 data are, as expected, reduced relative to the values for the 2013 data (approximately 3.5 times smaller).

The narrow sense SNP heritability from the ascertained case-control schizophrenia GWAS is estimated as $h^2 = 0.37$. Taking adult population prevalence of schizophrenia to be $K = 0.01$ [66, 67] (but see also [68], for $K = 0.005$), and given that there are 51,900 cases and 71,675 controls in the study, so that the proportion of cases in the study is $P = 0.42$, the heritability on the liability scale for schizophrenia from Eq 39 is $\hat{h}_l^2 = 0.21$. For bipolar disorder, with $K = 0.005$ [58], 20,352 cases and 31,358 controls, $\hat{h}_l^2 = 0.16$. Major depressive disorder appears to have a much lower model-estimated SNP heritability than schizophrenia: $\hat{h}_l^2 = 0.07$. The model estimate of SNP heritability for height is 17%, lower than the oft-reported value $\simeq$50% (see Discussion). However, despite the huge differences in sample size, we find the same value, 17%, for the 2010 GWAS ($N = 133,735$ [69]), and 19% for the 2018 GWAS ($N = 707,868$ [46, 49])—see Table 2.

Fig 3 shows the sample size required so that a given proportion of chip heritability is captured by genome-wide significant SNPs for the phenotypes (assuming equal numbers of cases and controls for the qualitative phenotypes: $N_{eff} = 4/(1/N_{cases} + 1/N_{controls})$, so that when $N_{cases} = N_{controls}$, $N_{eff} = N_{cases} + N_{controls} = N$, the total sample size, allowing for a straightforward comparison with quantitative traits). At current sample sizes, only 4% of narrow-sense chip heritability is captured for schizophrenia and only 1% for bipolar disorder; using current methodologies, a sample size of $N_{eff} \sim 1$ million would be required to capture the preponderance of SNP heritability for these phenotypes. Major depressive disorder GWAS currently is greatly under-powered, as shown in Fig 3(A). For education, we predict that 3.5% of

**Table 2. Summary of model results for phenotypes shown in Figs 1 and 2.** The subscript in $h_{(l)}^2$ indicates that for the qualitative phenotypes (the first eight) the reported SNP heritability is on the liability scale. MDD: Major Depressive Disorder; CAD: coronary artery disease; AD: Alzheimer's Disease (excluding APOE locus; *for the full autosomal reference panel, i.e., including APOE, $\hat{h}_l^2 = 0.15$ for AD—see S1 Appendix (p. S17)); BMI: body mass index; †ALS: amyotrophic lateral sclerosis, restricted to chromosome 9; LDL: low-density lipoproteins; HDL: high-density lipoproteins. $In addition to the 2014 height GWAS ($N = 251,747$ [46]), we include here model results for the 2010 ($N = 133,735$ [69]) and 2018 ($N = 707,868$ [49]) height GWAS; there is remarkable consistency for the 2010 and 2014 GWAS despite very large differences in the sample sizes—see S1 Appendix (p. S17). Confidence intervals are in S1 Appendix (p. S12).

| Phenotype | $\pi_1$ | $\sigma_\beta^2$ | $\sigma_0^2$ | $n_{causal}$ | $h_{(l)}^2$ |
|---|---|---|---|---|---|
| MDD | 4.01E-3 | 7.20E-6 | 1.06 | 4.4E4 | 0.07 |
| Bipolar Disorder | 2.70E-3 | 5.25E-5 | 1.05 | 3.0E4 | 0.16 |
| Schizophrenia | 2.84E-3 | 5.51E-5 | 1.14 | 3.1E4 | 0.21 |
| CAD | 1.14E-4 | 1.47E-4 | 0.97 | 1.3E3 | 0.03 |
| Ulcerative Colitis | 1.26E-4 | 8.82E-4 | 1.14 | 1.4E3 | 0.11 |
| Crohn's Disease | 9.56E-5 | 1.70E-3 | 1.17 | 1.1E3 | 0.18 |
| AD (no APOE)* | 1.11E-4 | 2.22E-4 | 1.05 | 1.2E3 | 0.08 |
| ALS† | 1.43E-5 | 3.04E-3 | 1.02 | 7 | 0.00 |
| Education | 3.20E-3 | 1.57E-5 | 1.00 | 3.5E4 | 0.12 |
| Intelligence | 2.20E-3 | 2.32E-5 | 1.28 | 2.4E4 | 0.13 |
| BMI | 6.44E-4 | 4.28E-5 | 0.88 | 7.5E3 | 0.07 |
| Height (2010)$ | 4.32E-4 | 1.66E-4 | 0.94 | 4.8E3 | 0.17 |
| Height (2014) | 5.66E-4 | 1.23E-4 | 1.66 | 6.2E3 | 0.17 |
| Height (2018)$ | 8.56E-4 | 9.46E-5 | 2.50 | 9.4E3 | 0.19 |
| Putamen Volume | 4.94E-5 | 9.72E-4 | 1.00 | 540 | 0.11 |
| LDL | 3.58E-5 | 6.61E-4 | 0.96 | 390 | 0.06 |
| HDL | 2.37E-5 | 1.25E-3 | 0.97 | 260 | 0.07 |
| TC | 4.26E-5 | 8.99E-4 | 0.96 | 469 | 0.09 |

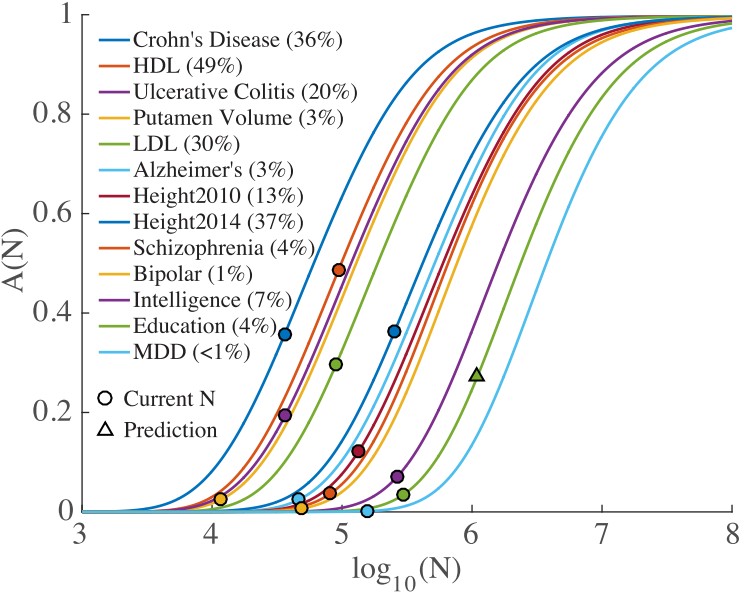

**Fig 3. Proportion of narrow-sense chip heritability, $A(N)$ (Eq 37), captured by genome-wide significant SNPs as a function of sample size, $N$, for phenotypes shown in Figs 1 and Fig 2.** Values for current sample sizes are shown in parentheses. Left-to-right curve order is determined by decreasing $\sigma_\beta^2$. The prediction for education at sample size N = 1.1 million is $A(N) = 0.27$, so that the proportion of phenotypic variance explained is predicted to be 3.5%, in good agreement with 3.2% reported in [70]. (The curve for AD excludes the APOE locus. For HDL, see S1 Appendix (p. S9) for additional notes).

phenotypic variance would be explained at $N$ = 1.1 million, in good agreement with the value found from direct computation of 3.2% [70]. For other phenotypes, the proportions of total SNP heritability captured at the available sample sizes are given in Fig 3.

The sample size for ALS was quite low, and we restricted the analysis to chromosome 9, which had most of the genome-wide significant typed SNPs; we estimate that there are $\simeq 7$ causal SNPs with high discoverability on chromosome 9 [71, 72], with very high discoverability, $\sigma_\beta^2 \simeq 0.003$. In contrast, for AD restricted to chromosome 19, there were an estimated 14 causal SNPs with discoverability $\sigma_\beta^2 \simeq 0.02$ (see the S1 Appendix (p. S17)).

In this study, we assume that population stratification in the raw data has been corrected for in the publicly-available summary statistics. However, given that some of the sample sizes are extremely large, we allow for the possibility of residual cryptic relatedness. This would result in a scaling of the z-scores, Eq 9 [19]. Thus, to test the modeling of inflation due to cryptic relatedness, we scaled the simulation z-scores as described earlier ($z = \sigma_0 z_u$ with $\sigma_0 > 1$, where $z_u$ are the original z-scores, i.e., not artificially inflated) and reran the model. E.g., for education and schizophrenia we inflated the z-scores by a factor of 1.2. For schizophrenia we found $\sigma_0^2 = 1.366$, which is almost exactly as predicted ($1.14 \times 1.2 = 1.368$), while the polygenicity and discoverability parameters are essentially unchanged: $\pi_1 = 2.81 \times 10^{-3}$, and $\sigma_\beta^2 = 5.56 \times 10^{-5}$. For education we found $\sigma_0^2 = 1.206$, which again is almost exactly as predicted ($1.0 \times 1.2 = 1.2$), while the polygenicity and discoverability parameters are again essentially unchanged: $\pi_1 = 3.19 \times 10^{-3}$, and $\sigma_\beta^2 = 1.57 \times 10^{-5}$.

A comparison of our results with those of [14] and [15] is in the S1 Appendix (p. S13). Critical methodological differences with model M2 in [14] are that we use a full reference panel of 11 million SNPs from 1000 Genomes Phase 3, we allow for the possibility of inflation in the data, and we provide an exact solution, based on Fourier Transforms, for the z-score pdf

arising from the posited distribution of causal effects, resulting in better fits of the model and the data QQ plots—as can be seen by comparing our QQ plots with those reported in S1 Appendix (p. S13). Although our estimated number of causal are often within a factor of two of those from the nominally equivalent model M2 of Zhang et al, there is no clear pattern to the mismatch.

## GWAS replication

In the S1 Appendix (pp. S7-S11) we provide an extensive example of testing the compatibility of summary statistics from two large bipolar disorder GWASs. Because z-scores are so noisy, it is possible for a typed SNP with a highly significant p-values in one GWAS to completely fail to reach significance in a subsequent GWAS, and for these outcomes to be statistically consistent. SNP heterozygosity and total LD, as well as sample sizes, are relevant in making such assessments.

## Dependence on reference panel

Given a liberal MAF threshold of 0.002, our reference panel should contain the vast majority of common SNPs for European ancestry. However, it does not include other structural variants (such as small insertions/deletions, or haplotype blocks) which may also be causal for phenotypes. To validate our parameter estimates for an incomplete reference, we reran our model on real phenotypes using the culled reference where we exclude every other SNP. The result is that all estimated parameters are as before except that $\hat{\pi}_1$ doubles, leaving the estimatde number of causal SNPs and heritability as before. For example, for schizophrenia we get $\pi_1 = 5.3 \times 10^{-3}$ and $\sigma_\beta^2 = 5.8 \times 10^{-5}$ for the reduced reference panel, versus $\pi_1 = 2.8 \times 10^{-3}$ and $\sigma_\beta^2 = 5.5 \times 10^{-5}$ for the full panel, with heritability remaining essentially the same (37% on the observed scale).

## Discussion

Here we present a unified method based on GWAS summary statistics, incorporating detailed LD structure from an underlying reference panel of SNPs with MAF>0.002, for estimating: phenotypic polygenicity, $\pi_1$, expressed as the fraction of the reference panel SNPs that have a non-null true $\beta$ value, i.e., are "causal"; and SNP discoverability or mean strength of association (the variance of the underlying causal effects), $\sigma_\beta^2$. In addition the model can be used to estimate residual inflation of the association statistics due to variance distortion induced by cryptic relatedness, $\sigma_0^2$. The model assumes that there is very little, if any, inflation in the GWAS summary statistics due to population stratification (bias shift in z-scores due to ethnic variation).

We apply the model to sixteen diverse phenotypes, eight qualitative and eight quantitative. From the estimated model parameters we also estimate the number of causal common-SNPs in the underlying reference panel, $n_{causal}$, and the narrow-sense common-SNP heritability, $h^2$ (for qualitative phenotypes, we re-express this as the proportion of population variance in disease liability, $h_l^2$, under a liability threshold model, adjusted for ascertainment); in the event rare SNPs (i.e., not in the reference panel) are causal, $h^2$ will be an underestimate of the true SNP heritability. In addition, we estimate the proportion of SNP heritability captured by genome-wide significant SNPs at current sample sizes, and predict future sample sizes needed to explain the preponderance of SNP heritability.

We find that schizophrenia is highly polygenic, with $\pi_1 = 2.8 \times 10^{-3}$. This leads to an estimate of $n_{causal} \simeq 31,000$, which is in reasonable agreement with a recent estimate that the

number of causal SNPs is >20,000 [73]. The SNP associations, however, are characterized by a narrow distribution, $\sigma_\beta^2 = 6.27 \times 10^{-5}$, indicating that most associations are of weak effect, i.e., have low discoverability. Bipolar disorder has similar parameters. The smaller sample size for bipolar disorder has led to fewer SNP discoveries compared with schizophrenia. However, from Fig 3, sample sizes for bipolar disorder are approaching a range where rapid increase in discoveries becomes possible. For educational attainment [42, 74, 75], the polygenicity is somewhat greater, $\pi_1 = 3.2 \times 10^{-3}$, leading to an estimate of $n_{causal} \simeq 35,000$, half a recent estimate, $\simeq 70,000$, for the number of loci contributing to heritability [74]. The variance of the distribution for causal effect sizes is a quarter that of schizophrenia, indicating lower discoverability. Intelligence, a related phenotype [43, 76], has a larger discoverability than education while having lower polygenicity ($\sim 10,000$ fewer causal SNPs).

In marked contrast are the lipoproteins and putamen volume which have very low polygenicity: $\pi_1 < 5 \times 10^{-5}$, so that only 250 to 550 SNPs (out of $\simeq 11$ million) are estimated to be causal. However, causal SNPs for putamen volume and HDL appear to be characterized by relatively high discoverability, respectively 17-times and 23-times larger than for schizophrenia (see S1 Appendix (p. S9) for additional notes on HDL, and [77] for a relevant comparison with our work).

The QQ plots (which are sample size dependent) reflect these differences in genetic architecture. For example, the early departure of the schizophrenia QQ plot from the null line indicates its high polygenicity, while the steep rise for putamen volume after its departure corresponds to its high SNP discoverability.

For Alzheimer's disease, our estimate of the liability-scale SNP heritability for the full 2013 dataset [39] is 15% for prevalence of 14% for those aged 71 older, half from APOE, while the recent "M2" and "M3" models of Zhang et al [14] gave values of 7% and 10% respectively–see S1 Appendix (p. S13). A recent report from two methods, LD Score Regression (LDSC) and SumHer [77], estimated SNP heritability of 3% for LDSC and 12% for SumHer (assuming prevalence of 7.5%). A raw genotype-based analysis (GCTA), including genes that contain rare variants that affect risk for AD, reported SNP heritability of 53% [7, 78]; an earlier related study that did not include rare variants and had only a quarter of the common variants estimated SNP heritability of 33% for prevalence of 13% [79]. GCTA calculations of heritability are within the domain of the so-called infinitesimal model where all markers are assumed to be causal. Our model suggests, however, that phenotypes are characterized by polygenicities less than $5 \times 10^{-3}$; for AD the polygenicity is $\simeq 10^{-4}$. Nevertheless, the GCTA approach yields a heritability estimate closer to the twin-based (broad sense) value, estimated to be in the range 60-80% [80]. The methodology appears to be robust to many assumptions about the distribution of effect sizes [81, 82]; the SNP heritability estimate is unbiased, though it has larger standard error than methods that allow for only sparse causal effects [69, 83]. For the 2013 data analyzed here [39], a summary-statistics-based method applied to a subset of 54,162 of the 74,046 samples gave SNP heritability of almost 7% on the observed scale [12, 84]; our estimate is 12% on the observed scale—see S1 Appendix (p. S17).

Onset and clinical progression of sporadic Alzheimer's disease is strongly age-related [85, 86], with prevalence in differential age groups increasing at least up through the early 90s [62]. Thus, it would be more accurate to assess heritability (and its components, polygenicity and discoverability) with respect to, say, five-year age groups beginning with age 65 years, and using a consistent control group of nonagenarians and centenarians. By the same token, comparisons among current and past AD GWAS are complicated because of potential differences in the age distributions of the respective case and the control cohorts. Additionally, the degree to which rare variants are included will affect heritability estimates. The summary-statistic-

based estimates of polygenicity that we report here are, however, likely to be robust for common SNPs: $\pi_1 \simeq 1.1 \times 10^{-4}$, with only a few causal SNPs on chromosome 19.

Our point estimate for the liability-scale SNP heritability of schizophrenia is $h_l^2 = 0.21$ (assuming a population risk of 0.01), and that 4% of this (i.e., 1% of overall disease liability) is explainable based on common SNPs reaching genome-wide significance at the current sample size. This $h_l^2$ estimate is in reasonable agreement with a recent result, $h_l^2 = 0.27$ [73, 87], also calculated from the PGC2 data set but using raw genotype data for 472,178 markers for a subset of 22,177 schizophrenia cases and 27,629 controls of European ancestry; and with an earlier result of $h_l^2 = 0.23$ from PGC1 raw genotype data for 915,354 markers for 9,087 schizophrenia cases and 12,171 controls [7, 88]. The recent "M2" (single non-null Gaussian) model estimate is $h_l^2 = 0.29$ [14] (see S1 Appendix (p. S13)). No QQ plot was available for the M2 model fit to schizophrenia data, but such plots (truncated on the y-axis at $-\log_{10}(p) = 10$) for many other phenotypes were reported [14]. We note that for multiple phenotypes (height, LDL cholesterol, total cholesterol, years of schooling, Crohn's disease, coronary artery disease, and ulcerative colitis) our single causal Gaussian model appears to provide a better fit to the data than M2: many of the M2 plots show a very early and often dramatic deviation between prediction and data, as compared with our model QQ plots which are also built from a single causal Gaussian, suggesting an upward bias in polygenicity and/or variance of effect sizes, and hence heritability as measured by the M2 model for these phenotypes. The LDSC liability-scale (1% prevalence) SNP heritability for schizophrenia has been reported as $h_l^2 = 0.555$ [12] and more recently as 0.19 [77], the latter in very good agreement with our estimate; on the observed scale it has been reported as 45% [12, 84], in contrast to our corresponding value of 37%. Our estimate of 1% of overall variation on the liability scale for schizophrenia explainable by genome-wide significant loci compares reasonably with the proportion of variance on the liability scale explained by Risk Profile Scores (RPS) reported as 1.1% using the "MGS" sample as target (the median for all 40 leave-one-out target samples analyzed is 1.19%—see Extended Data Figure 5 and Supplementary Tables 5 and 6 in [36]; this was incorrectly reported as 3.4% in the main paper). These results show that current sample sizes need to increase substantially in order for RPSs to have predictive utility, as the vast majority of associated SNPs remain undiscovered. Our power estimates indicate that ~500,000 cases and an equal number of controls would be needed to identify these SNPs (note that there is a total of approximately 3 million cases in the US alone).

A subtle but important issue is downward bias of large-sample maximum-likelihood estimates of SNP heritability, due to over-ascertainment of cases in case-control studies [87]; it has been examined in the context of restricted maximum likelihood (REML) in GCTA, which assumes a polygenicity of 1, i.e., every SNP is causal. For schizophrenia, this has been assessed in the context of BOLT-REML, which assumes a mixture distribution of small ('spike') and large ('slab') effects [73]: from 22,177 cases and 27,629 controls, the observed-scale heritability is reported as $h^2 = 0.415$, equivalent to $h_l^2 = 0.23$ on the liability scale, assuming 1% disease prevalence. However, using "phenotype correlation-genetic correlation" (PCGC) regression, a moments-based approach requiring raw-genotype data which produces unbiased estimates for case-control studies of disease traits [87], the unbiased liability-scale heritability is reported as $h_g^2 = 0.27$, indicating that the likelihood-maximization estimate is biased down by 15% of the unbiased value (the degree of underestimation decreases for smaller sample sizes). Our estimate for the liability-scale heritability of schizophrenia, from a larger sample than in [73], is $h_l^2 = 0.21$. This at least would be consistent with downward bias operating in point-normal causal distributions, in a manner similar to that in GCTA and BOLT-REML. This would then translate into either an underestimate of the number of causal SNPs, or more likely an underestimate of the variance of the distribution of causal effects.

For educational attainment, we estimate SNP heritability $h^2 = 0.12$, in good agreement with the estimate of 11.5% given in [42]. As with schizophrenia, this is substantially less than the estimate of heritability from twin and family studies of $\simeq$40% of the variance in educational attainment explained by genetic factors [74, 89].

For putamen volume, we estimate the SNP heritability $h^2 = 0.11$, in reasonable agreement with an earlier estimate of 0.1 for the same overall data set [4, 47]. For LDL and HDL, we estimate $h^2 = 0.06$ and $h^2 = 0.07$ respectively, in good agreement with the LDSC estimates $h^2 = 0.08$ and $h^2 = 0.07$ [77], and the M2 model of [14]—see S1 Appendix (p. S13).

For height (N = 251,747 [46]) we find that its model polygenicity is $\pi_1 = 5.66 \times 10^{-4}$, a quarter that of intelligence, while its discoverability is five times that of intelligence, leading to a SNP heritability of 17%. The number of causal SNPs (out of a total of approximately 11 million) is approximately 6k; although this is about one twentieth the estimate reported in [90], it remains large and allows for height to be interpreted as "omnigenic". For the 2010 GWAS (N = 133,735 [69]) and 2018 GWAS (N = 707,868 [49]), we estimate SNP heritability of 17% and 19% respectively (see Table 2, and S1 Appendix (p. S17)). These heritabilities are in considerable disagreement with the SNP heritability estimate of $\simeq$50% [46] (average of estimates from five cohorts ranging in size from N = 1,145 to N = 5,668, with $\simeq$1 million SNPs). For the 2010 GWAS, the M2 model [14] gives $h^2 = 0.30$ (see S1 Appendix (p. S13)); the upward deviation of the model QQ plot in [14] suggests that this value might be inflated. For the 2014 GWAS, the M3 model estimate is $h^2 = 33\%$ [14]; the Regression with Summary Statistics (RSS) model estimate is $h^2 = 52\%$ (with $\simeq$11, 000 causal SNPs) [91], which, not taking any inflation into account, is definitely a model overestimate; and in [77] the LDSC estimate is reported as $h^2 = 20\%$ while the SumHer estimate is $h^2 = 46\%$ (in general across traits, the SumHer heritability estimates tend to be two-to-five times larger than the LDSC estimates). The M2, M3, and RSS models use a reference panel of $\simeq$1 million common SNPs, in contrast with the $\simeq$11 million SNPs used in our analysis. Also, it should be noted that the M2, M3, and RSS model estimates did not take the possibility of inflation into account. For the 2014 height GWAS, that inflation is reported as the LDSC intercept is 2.09 in [77], indicating considerable inflation; for the 2018 dataset we find $\sigma_0^2 = 2.5$, while the LD score regression intercept is 2.1116 (se 0.0458). Given the various estimates of inflation and the controversy over population structure in the height data [50, 51], it is not clear what results are definitely incorrect.

Our power analysis for height (2014) shows that 37% of the narrow-sense heritability arising from common SNPs is explained by genome-wide significant SNPs ($p \leq 5 \times 10^{-8}$), i.e., 6.3% of total phenotypic variance, which is substantially less than the 16% direct estimate from significant SNPs [46]. It is not clear why these large discrepancies exist. One relevant factor, however, is that we estimate a considerable confounding ($\sigma_0^2 = 1.66$) in the height 2014 dataset. Our $h^2$ estimates are adjusted for the potential confounding measured by $\sigma_0^2$, and thus they represent what is likely a lower bound of the actual SNP-heritability, leading to a more conservative estimate than what has previously been reported. We note that after adjustment, our $h^2$ estimates are consistent across all three datasets (height 2010, 2014 and 2018), which otherwise would range by more than 2.5-fold. Another factor might be the relative dearth of typed SNPs with low heterozygosity and low total LD (see top left segment in S1 Appendix (p. S29), $n = 780$): there might be many causal variants with weak effect that are only weakly tagged. Nevertheless, given the discrepancies noted above, caution is warranted in interpreting our model results for height.

## Conclusion

The common-SNP causal effects model we have presented is based on GWAS summary statistics and detailed LD structure of an underlying reference panel, and assumes a Gaussian

distribution of effect sizes at a fraction of SNPs randomly distributed across the autosomal genome. While not incorporating the effects of rare SNPs, we have shown that it captures the broad genetic architecture of diverse complex traits, where polygenicities and the variance of the effect sizes range over orders of magnitude.

The current model (essentially Eq 4) and its implementation (essentially Eq 29) are basic elements for building a more refined model of SNP effects using summary statistics. Higher accuracy in characterizing causal alleles in turn will enable greater power for SNP discovery and phenotypic prediction.

## Supporting information

**S1 Appendix. Appendix.** Additional text, tables, and figures.
(PDF)

**S1 DataSourceList. Data source list.**
(XLSX)

## Acknowledgments

We thank the consortia for making available their GWAS summary statistics, and the many people who provided DNA samples.

## Author Contributions

**Conceptualization:** Dominic Holland, Anders M. Dale.

**Data curation:** Oleksandr Frei.

**Formal analysis:** Dominic Holland.

**Funding acquisition:** Ole A. Andreassen, Anders M. Dale.

**Investigation:** Dominic Holland, Oleksandr Frei, Ole A. Andreassen, Anders M. Dale.

**Methodology:** Dominic Holland, Anders M. Dale.

**Software:** Dominic Holland.

**Validation:** Dominic Holland.

**Writing – original draft:** Dominic Holland.

**Writing – review & editing:** Dominic Holland, Oleksandr Frei, Rahul Desikan, Chun-Chieh Fan, Alexey A. Shadrin, Olav B. Smeland, V. S. Sundar, Paul Thompson, Ole A. Andreassen.

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
