## [Decision Letter · Decision Letter 0]

5 Aug 2019

Dear Dr Holland,

Thank you very much for submitting your Research Article entitled 'Beyond SNP Heritability: Polygenicity and Discoverability of Phenotypes Estimated with a Univariate Gaussian Mixture Model' to PLOS Genetics. Your manuscript was fully evaluated at the editorial level and by independent peer reviewers. The reviewers appreciated the attention to an important problem, but raised some substantial concerns about the current manuscript. Based on the reviews, we will not be able to accept this version of the manuscript, but we would be willing to review again a much-revised version. We cannot, of course, promise publication at that time.

In particular, please provide the software codes and the detailed simulations, as all three reviewers have a question about the availability of the software.

If you decide to revise the manuscript for further consideration at PLOS Genetics, please aim to resubmit within the next 60 days, unless it will take extra time to address the concerns of the reviewers, in which case we would appreciate an expected resubmission date by email to plosgenetics@plos.org.

[LINK]

We are sorry that we cannot be more positive about your manuscript at this stage. Please do not hesitate to contact us if you have any concerns or questions.

Yours sincerely,

Xiaofeng Zhu

Associate Editor

PLOS Genetics

Scott Williams

Section Editor: Natural Variation

PLOS Genetics

Reviewer's Responses to Questions

**Comments to the Authors:**

Reviewer #1: In this report, the authors proposed a general computitional method that can infer the genetic architecture underlying complex traits. an idea similar to LD score regression, using a mixture Gaussian distribution GWAS summary statistics have been combined with a reference population (1KG); a powerful method, as demonstrated by the authors, various interesting parameters and questions, such as the proportion of causal variants and their variance can be addressed. In real GWAS summary data analysis, a range of GWAS complex traits have been tested and consistent to previous reported results in other studies.

Technical comments

1 Similar ideas but using individual-level data had been proposed in Bayesian approaches [Nat Genet, Zeng, 2018]. In particular, their model assume the effect size is relavent to MAF. It is likely to be the case that the MAF and effect has certain correlation, negative or positive upon the scheme of the selection pattern. It is probably one of the reasons that some z-score distribution was not fitted as well as others in the manuscript.

2 The kernel of the method is the introduction of the mixture model that is composed of the causal and null components. In the multinomial pdf function Eq 29 in Supplementary notes, H_i is taken as the mean 2pq for the n_i snps in the bin. However, as k_i is a small number, say k_i =1, the 2pq of this specific causal locus will be very different from the mean 2pq of n_i snps in this bin. It is not really clear H_i in Eq 29 has been updated for k_i or always taken as H_i as approximation? In gut feeling, it may effect the estimation. However, the authors emphized that the real calculation was based on convolution,

3 Eq 51. H_bar is taken as 0.2165, similar critics as in the last comment.

4 The LD score is estimated from the 1KG, which is presumed to be “health controls”. For case-control traits, the LD in the case-control sample will be different from “health controls”, in particular the effect size is large or the trait in under selection. Any consequence due to this leap?

5 The estimation procedure should be better demonstrated in the main text, which is the kernel of the paper. As well as its step-by-step algorithm, and demo code should be provided for others to follow up.

Minors

1 The main text and the supplementary notes are very similar to each other.

2 References are inconsistent between the main and the supplementary notes.

3 Many typos. The ms has not been lined well for the reviewer to tell where those typos. In the supplementary materials it is also referred to “Supplementary Materials”, page 16 second column.

4 Probably the author is from physics field, I guess. The authors used some jargons such as Dirac delta function. However, according the sup Notes, Eq 6-8, the Dirac delta function seems a standard normal distribution instead.

Reviewer #2: In this paper the authors present a method for estimating the genetic architecture of a complex trait from the set of GWAS summary statistics and an estimate of the LD matrix. From a methodological perspective, the major contribution of the paper seems to be the methods, presented in the supplement, for fitting the model using summary statistics and the alone. In the model, the marginal effect sizes for each individiaul SNP are assumed to be generated due to tagging effects of multiple linked SNPs, where the causal effects of the linked SNPs follow a mixture distribution that is a combination of a point mass at zero, and a normal distribution with some variance. Such mixture distributions are fairly common in the field, but one contribution here seems to be a conceptual stand on principle that a significant number of variants should have no effect on a given trait at all, and the methods here fit such a model (whereas most mixture distribution approaches assume that all sites makes at least some small contribution).

In terms of the presentation, I think it would be helpful if the authors could present a bit more of the details of the model could be presented. E.g. I see no reason why equations 5 and 16 from the supplement (along with perhaps whatever others the authors feel necessary to provide appropriate context; see the following) couldn't appear in the main text. More broadly, I think the paper could do more to help place the present work in the context of existing methods of a similar flavor (of which there are many). What specifically distinguishes the approaches laid out here from those which can be found elsewhere in the literature. The reader shouldn't have to take a detour to read 10 other paper in order to figure it out.

I must confess I am a bit concerned about identifiability issues here. E.g. the authors show in the supplement that if you simulate data under diffiferent assumptions, you can somewhat seriously misestimate the number of causal SNPs. In that specific case (where effects come from a particular mixture of Gaussians), the show that their model does not fit the QQ plot well, so we might take the fact that the the model does fit the QQ plots for real traits as a reason to believe in the estimates, but this is weak evidence at best. In the one case where the authors have real data for the same trait at multiple different samples sizes (height), the estimated number of causal SNPs is correlated with GWAS sample size. It is of course hard to know exactly what to make of this, given the known structure confounding issues in the GIANT consortium data, but it is not super reassuring. If the true effect size distribution exhibits the sort of long flat tails the emerge from scale mixtures of Gaussians (i.e. the sort that is explored in the supplement), I would not be surprised to see sample size dependence of parameter estimates, particularly the polygenicity, if the effect sizes truly follow this distribution.

In general, I don't like to suggest too much extra work in review, but it strikes me that it would be reassuring to see something like the UK biobank height data subsetted to different sample sizes, as a demonstration that you get consistent estimates of the number of causal sites. I.e. if you run 10 GWAS on resampled subsets of 50k UK Biobank participants, and then apply your method, do you get roughly the same average estimates for the parameters as you do if you run 10 GWAS on different subsets of 200k individuals?

I also wanted to respond briefly to note that, while it is a very minor point in the context of this paper, I think the authors have a mistaken view of what the omnigenic model is. The omnigenic model, as articulated by Boyle et al, and more recently by Liu et al, is not a claim about the total number of casual SNPs that exist for complex phenotypes. Rather, it is a conceptual model meant to explain multiple features that are commonly observed in GWAS data. One of these features is that the total number of causal SNPs is high, but I think those authors would argue that ~10k is still "high", as far as they are concerned. But another important feature is the fact that much heritability seems to be attributable to variants within genes that are not directly related to the the trait in question, but simply expressed in the relevant cell type. In fact, the "omni" in "omnigenic" is best interpretted in this context. It means roughly "all genes expressed in the relevant cell types affect the trait" not "all variants affect all traits", which is what I read the authors to be arguing against when they say in the main text that height is "very far from omnigenic".

Is software to fit the model made available? I could not find any mention of it in the text.

Reviewer #3: This manuscript describes a new method for estimating heritability and the number of causal SNPs for GWAS traits. The proposed method relies on a spike-slab model and perform a grid search based on MAF and LD pattern for inference. The authors applied the model to analyze 16 diverse phenotypes including eight binary disease traits. Overall, the manuscript is well written, and the analysis is well carried out. The propose algorithm makes sense and appears to work well. With that said, however, I do have a few main concerns:

1. The proposed spike-slab model is identical to that used in ref #14 and one of the proposed inference algorithms (the one based on the multinomial expansion) is also very similar to the one proposed in ref #14. Therefore, it would be important to provide a detailed description of the method in ref #14 and explain how the present method differs from that of ref #14. In addition, it would be important to perform comprehensive comparisons with the ref #14 method in all simulations and real data applications, to demonstrate any benefits of the proposed method.

2. The authors did not provide any software. Some of the proposed algorithms are complicated. Without providing an open-source and user-friendly software along with a software manual, it would be almost impossible for anyone to use the proposed method.

3. The authors did not provide the analysis code used in the paper. I am thus unable to reproduce any of their simulation and real data analysis results. It would be important to provide all analysis code along with all analyzed gwas summary statistics.

4. It has been shown previously that sparse models, like the spike-slab model used in the present paper, are not suitable for heritability estimation under polygenic settings. Indeed, sparse models often under-estimate heritability, at least in studies with small samples (e.g. as illustrated in ref #71). I am thus surprised to see the h2 estimates are fine in the simulations. I was wondering if this is due to the unrealistic simulations employed in the present paper: all simulations were based on simulated SNPs but not based on SNPs from real data. Therefore, it would be important to perform realistic simulations either using real genotype data or following the simulation approach used in ref #14.

5. Half of the analyzed traits in the present study are binary disease traits. It is well known that likelihood-based approaches are not suitable for h2 estimation for case control studies, as likelihood-based method would bias the estimation of h2 due to case control ascertainment (e.g. ref #75). I am thus concerned whether the proposed method can be used for analyzing binary traits. It would be important to provide mathematical details on how the proposed model can accommodate case control studies. For example, do you need to incorporate the spike-slab prior to the liability threshold model? Do you need to develop moment matching type of algorithms for inference? It would also be important to perform comprehensive simulations under ascertained case control settings, like those have been done in ref #75.

**Have all data underlying the figures and results presented in the manuscript been provided?**

Reviewer #1: No: demo code is not provided

Reviewer #2: Yes

Reviewer #3: No:

PLOS authors have the option to publish the peer review history of their article (what does this mean?). If published, this will include your full peer review and any attached files.

Reviewer #1: No

Reviewer #2: No

Reviewer #3: No

---

## [Decision Letter · Decision Letter 1]

5 Nov 2019

Dear Dr Holland,

Thank you very much for submitting your Research Article entitled 'Beyond SNP Heritability: Polygenicity and Discoverability of Phenotypes Estimated with a Univariate Gaussian Mixture Model' to PLOS Genetics. Your manuscript was fully evaluated at the editorial level and by independent peer reviewers. The reviewers appreciated the attention to an important problem, but raised some substantial concerns about the current manuscript. Based on the reviews, we will not be able to accept this version of the manuscript, but we would be willing to review again a much-revised version. We cannot, of course, promise publication at that time.

If you decide to revise the manuscript for further consideration at PLOS Genetics, please aim to resubmit within the next 60 days, unless it will take extra time to address the concerns of the reviewers, in which case we would appreciate an expected resubmission date by email to plosgenetics@plos.org.

[LINK]

We are sorry that we cannot be more positive about your manuscript at this stage. Please do not hesitate to contact us if you have any concerns or questions.

Yours sincerely,

Xiaofeng Zhu

Associate Editor

PLOS Genetics

Scott Williams

Section Editor: Natural Variation

PLOS Genetics

I suggested that the authors carefully address the reviewer 3's comments, which are also the previous comments 1 and 5 by the reviewer 3.

Reviewer's Responses to Questions

**Comments to the Authors:**

Reviewer #1: Congratulations!

It is very helpful to provide the code for the proposed analysis.

Reviewer #3: My previous main comments #1 and #5 were not well addressed. In the previous comment #1, I suggested the authors to perform a comprehensive comparison with previous methods developed in ref #14 (M2 and M3 methods) in all simulations and real data applications, due to the apparent similarity between the proposed method and the previous M2/M3 methods. The authors only criticized the previous M2/M3 methods being incorrect and applied M2/M3 models to real data. In the real data applications, the authors showed that the proposed method tends to produce a higher estimated number of causal SNPs than the previous M2/M3 methods and tends to estimate the heritability lower than the previous M2/M3 methods. Without simulations, however, it is unclear whether such difference observed in the real data is due to the inaccurate estimates from the proposed method or the inaccurate estimates from the previous M2/M3 methods.

In the previous comment #5, I suggested the authors to perform simulations based on the liability threshold model to examine any potential downward bias in heritability estimation for binary traits. The authors explained that the estimates from the proposed method did produce downward bias (in the case of schizophrenia, about 15% downward bias). Because the estimates from M2/M3 models for all binary traits are all higher than that from the proposed method, I was wondering if the previous M2/M3 methods were not susceptible to case control ascertainment and thus could produce unbiased heritability estimates. It would thus be important to perform simulations based on the liability threshold model, to better understand whether the biased heritability estimation is a unique drawback for the proposed method, or whether it also applies to M2/M3.

**Have all data underlying the figures and results presented in the manuscript been provided?**

Reviewer #1: Yes

Reviewer #3: Yes

PLOS authors have the option to publish the peer review history of their article (what does this mean?). If published, this will include your full peer review and any attached files.

Reviewer #1: No

Reviewer #3: No

---

## [Editor Report · Decision Letter 2]

15 Jan 2020

Dear Dr Holland,

We are pleased to inform you that your manuscript entitled "Beyond SNP Heritability: Polygenicity and Discoverability of Phenotypes Estimated with a Univariate Gaussian Mixture Model" has been editorially accepted for publication in PLOS Genetics. Congratulations!

Yours sincerely,

Xiaofeng Zhu

Associate Editor

PLOS Genetics

Scott Williams

Section Editor: Natural Variation

PLOS Genetics

Comments from the reviewers (if applicable):

**Data Deposition**

http://datadryad.org/submit?journalID=pgenetics&manu=PGENETICS-D-19-00865R2

**Press Queries**

---

## [Editor Report · Acceptance letter]

27 Apr 2020

PGENETICS-D-19-00865R2 

Beyond SNP Heritability: Polygenicity and Discoverability of Phenotypes Estimated with a Univariate Gaussian Mixture Model 

Dear Dr Holland, 

We are pleased to inform you that your manuscript entitled "Beyond SNP Heritability: Polygenicity and Discoverability of Phenotypes Estimated with a Univariate Gaussian Mixture Model" has been formally accepted for publication in PLOS Genetics! Your manuscript is now with our production department and you will be notified of the publication date in due course.

With kind regards,

Kaitlin Butler

PLOS Genetics

On behalf of:
